# Overcoming global inequality is critical for land-based mitigation in line with the Paris Agreement

Florian Humpenöder [1]✉, Alexander Popp [1], Carl-Friedrich Schleussner [2,3], Anton Orlov[4], Michael Gregory Windisch[1,3], Inga Menke [2,3], Julia Pongratz[5,6], Felix Havermann[5], Wim Thiery [7], Fei Luo [8,9], Patrick v. Jeetze [1,3], Jan Philipp Dietrich [1], Hermann Lotze-Campen [1,3], Isabelle Weindl [1] & Quentin Lejeune [2]

Transformation pathways for the land sector in line with the Paris Agreement depend on the assumption of globally implemented greenhouse gas (GHG) emission pricing, and in some cases also on inclusive socio-economic development and sustainable land-use practices. In such pathways, the majority of GHG emission reductions in the land system is expected to come from low- and middle-income countries, which currently account for a large share of emissions from agriculture, forestry and other land use (AFOLU). However, in low- and middle-income countries the economic, financial and institutional barriers for such transformative changes are high. Here, we show that if sustainable development in the land sector remained highly unequal and limited to high-income countries only, global AFOLU emissions would remain substantial throughout the 21st century. Our model-based projections highlight that overcoming global inequality is critical for land-based mitigation in line with the Paris Agreement. While also a scenario purely based on either global GHG emission pricing or on inclusive socio-economic development would achieve the stringent emissions reductions required, only the latter ensures major co-benefits for other Sustainable Development Goals, especially in low- and middle-income regions.

In the period 2010–2019, global GHG emissions from the land sector (AFOLU) accounted for 13–21% of global total net anthropogenic GHG emissions, according to IPCC AR6 Working Group III[1]. GHG emissions from AFOLU consist of two major components: (a) carbon dioxide ($CO_2$) emissions and removals from land-use change and management, and (b) methane ($CH_4$) and nitrous oxide ($N_2O$) emissions from agriculture[2]. In 2018, $CO_2$ emissions from deforestation and other land conversions together with carbon uptake due to regrowth and re/

afforestation accounted for 47% of global net AFOLU GHG emissions, followed by $CH_4$ emissions from enteric fermentation with 25%[2]. Smaller sources of AFOLU GHG emissions include managed soils ($N_2O$), rice cultivation ($CH_4$), manure management ($CH_4$ and $N_2O$), and synthetic fertilizer application ($N_2O$). Unlike other sectors such as energy, industry, and transport, the relative contribution of AFOLU to overall GHG emissions is typically higher in developing than in developed countries[2]. In Africa, Latin America, and Southeast Asia, AFOLU

[1]Potsdam Institute for Climate Impact Research (PIK), Member of the Leibniz Association, Potsdam, Germany. [2]Climate Analytics (CA), Berlin, Germany. [3]Humboldt University of Berlin, Berlin, Germany. [4]CICERO, Oslo, Norway. [5]Ludwig-Maximilians-University (LMU) Munich, Munich, Germany. [6]Max Planck Institute for Meteorology, Hamburg, Germany. [7]Vrije Universiteit Brussel, Brussels, Belgium. [8]Institute for Environmental Studies, Vrije Universiteit Amsterdam, Amsterdam, The Netherlands. [9]Royal Netherlands Meteorological Institute (KNMI), De Bilt, The Netherlands. ✉e-mail: humpenoeder@pik-potsdam.de

GHG emissions account for >50% of total GHG emissions, mostly due to high $CO_2$ emissions from land-use change and management[1,2]. In contrast, the share of AFOLU in total GHG emissions is only 7% each in Europe and North America[1,2]. In absolute terms, AFOLU GHG emissions in 2015 were considerably higher in developing countries (9.5 Gt $CO_2$ eq) compared to industrialized countries (2.6 Gt $CO_2$ eq)[3]. Thus, low- and middle-income countries play an important role in AFOLU GHG emission reductions in transformation pathways in line with the Paris Agreement. In particular, $CH_4$ emissions from enteric fermentation are considerably lower in Africa and Asia in pathways that limit the global mean temperature increase to 1.5 °C above pre-industrial levels compared to scenarios with 3–4 °C warming[1]. Moreover, net $CO_2$ emissions from land-use change and management in Africa and Latin America are not only considerably reduced in 1.5 °C pathways but turn negative due to increased re- and afforestation[1].

To a large extent, GHG emission reductions in such transformation pathways depend on the assumption of a globally coordinated GHG emission pricing scheme across different sectors including energy, industry, transport, buildings, and AFOLU[4,5]. Some transformation pathways go beyond meeting the Paris Agreement climate objectives[6] and aim to maximize co-benefits with the broader United Nations Sustainable Development Goals (SDG) agenda[7,8]. In such sustainable transformation pathways, global GHG emission pricing is often complemented with global-scale inclusive socio-economic development (a convergence of all countries towards, e.g., the same demographic profiles, lower food waste levels, and healthy diets) and sustainable land-use practices (e.g., ecosystem protection and efficiency improvements)[7,8]. These assumptions facilitate further AFOLU GHG emissions reductions in sustainable transformation pathways, especially in low- and middle-income countries. However, there are considerable economic, financial, technical, and institutional barriers, especially in low- and middle-income countries, to implementing AFOLU GHG mitigation measures and making progress in key SDG areas such as health, education, and food security[9–11]. These barriers could result in a world of deepening inequalities where effective AFOLU GHG emission regulation and sustainable land-use practices remain limited to high-income countries only. At the same time, failure to make progress in key SDG areas might result in highly unequal socio-economic development within low- and middle-income countries and especially compared to high-income countries. For instance, access to education is a key driver for lowering population growth[12], which in turn is a key driver for future food demand and agricultural land use[13,14]. Thus, limited access to education in a world of global inequality (shared socio-economic pathway 4; SSP4) could result in far higher population growth, especially in Africa, compared to a world of global sustainable development (SSP1)[15]. Also, the impacts of the COVID-19 pandemic on the global economy will most likely further slow down progress across SDGs[16], especially in low-income countries in Sub-Saharan Africa[17]. Moreover, the Russian invasion of Ukraine might negatively affect food security, and in consequence political stability, in countries with limited coping capacity, especially in the Middle East and Africa[18]. In light of the closing window for reaching the global 1.5 °C target[11], it seems highly relevant to study how (in)compatible AFOLU GHG emissions in a world of deepening inequalities would be with the Paris Agreement climate objectives[6]. Likewise, a forward-looking analysis of expanding mitigation options from high-income regions to the global level can provide insights on the most effective options for AFOLU emission reduction in low- and middle-income regions under consideration of co-benefits and trade-offs with other SDGs. Such scenarios have not previously been developed and analyzed within a consistent modeling framework[5,8,13,14,19].

Here, we present a model-based quantification of two contrasting future scenarios for the AFOLU sector, complemented by three intermediate scenarios (Table 1; see Tables 2–4 and Methods for details). *Global-Inequality* is a scenario where sustainable development in the land sector remains highly unequal and limited to high-income countries only, whereas the scenario *Global-Sustainability* assumes global implementation of AFOLU GHG emission pricing, sustainable land-use practices, and inclusive socio-economic development. For the scenario quantification, we use the global multi-regional MAgPIE 4 open-source land-use modeling framework[20,21]. We find that a scenario of deepening inequalities with AFOLU GHG emission pricing in high-income countries only would not achieve the required reduction of global AFOLU GHG emissions for compatibility with the Paris Agreement. Extending AFOLU GHG emission pricing to the global level under unequal socio-economic development would achieve the stringent emission reductions required. However, global inclusive socio-economic development would synergically reduce global AFOLU GHG emissions in line with the 1.5 °C target and contribute to achieving multiple other SDGs.

## Results
### Scenario narratives
The two main scenarios, *Global-Inequality* and *Global-Sustainability*, differ in regional assumptions for three broad domains (Table 3): socio-economic development (population, income, diets, food waste; Figs. 1, 2), environmental protection (land, water, nitrogen) and land-based mitigation (GHG emission pricing, re/afforestation, bioenergy). The underlying narratives of both scenarios are the result of an iterative co-development process conducted between scientists with a focus on interactions between land use and climate on the one hand, and a group of land-use experts from academic and non-academic institutions on the other hand (see methods). *Global-Sustainability* is characterized by inclusive socio-economic development (lower population growth, reduced food waste, transition to EAT-Lancet planetary health diet[22]), sustainable land-use practices (biodiversity hotspot and environmental flow protection, improved fertilizer efficiency, conservation of a share of native habitats within working landscapes) and AFOLU GHG emission pricing ($CO_2$ emissions from land conversion, incentive for re/afforestation, non-$CO_2$ emissions from agriculture) in all world regions. In contrast, sustainable land-use practices and AFOLU GHG emission pricing remain limited to high-income regions (represented by OECD90 + EU in our study; Table 4) in *Global-Inequality*. Moreover, socio-economic development in low- and middle-income regions (Asia, Latin America, Sub-Saharan Africa, (MEA), reforming countries) strongly diverges from the development in high-income regions (e.g., higher population growth and higher prevalence of underweight; Fig. 1). These two main scenarios are complemented by three intermediate scenarios for decomposing the drivers for the transformation from *Global-Inequality* towards *Global-Sustainability* (Table 1): Global AFOLU GHG emission pricing (*Global-GHG Price*), global sustainable land-use practices (*Global-EnvirProt*) and global inclusive socio-economic development (*Global-SustDemand*). Some of our indicators can be mapped to the United Nations SDGs, which are aspirational goals for 2030. (Table S1): Prevalence of underweight (SDG 2: end hunger), prevalence of obesity (SDG 3: health), AFOLU GHG emissions (SDG 13: climate action), agricultural water use (SDG 6: clean water and sanitation), nitrogen fixation (SDG 15: life on land) and change of forest area (SDG 15).

### Synergies and trade-offs between land-based mitigation and other SDGs
This sub-section provides a summary of the main scenario results for the year 2050 with a focus on co-benefits and trade-offs between land-based mitigation and other SDGs (Fig. 3). Detailed results on land-use dynamics (Figs. 4, 5), AFOLU GHG emissions (Figs. 6, 7), and agricultural water use and nitrogen fixation (Fig. 8) are provided in the subsequent sub-sections. Numerical scenario results and raw data for figures are provided in Supplementary Datasets 1, 2, respectively.

**Table 1 | Summary of scenario definitions**

| Scenario name | AFOLU GHG emission pricing | Sustainable land-use practices | Socio-economic drivers and dietary change |
|---|---|---|---|
| Global-Inequality | OECD90 + EU | OECD90 + EU | SSP4 |
| Global-GHG Price | Global | OECD90 + EU | SSP4 |
| Global-EnvirProt | OECD90 + EU | Global | SSP4 |
| Global-SustDemand | OECD90 + EU | OECD90 + EU | SSP1 + EAT-Lancet diet + lower food waste |
| Global-Sustainability | Global | Global | SSP1 + EAT-Lancet diet + lower food waste |

Details and regional definitions are provided in Tables 2–4.

**Table 2 | Regional mapping**

| Name of region | Income group | MAgPIE region(s) |
|---|---|---|
| OECD90 + EU | high-income | Canada, Australia, New Zealand (CAZ), European Union (EUR), non-EU member states (NEU), United States (USA), Japan (JPN) |
| ASIA | middle-income | India (IND), China (CHA), other Asia (OAS) |
| LAM | middle-income | Latin America (LAM) |
| SSA | low-income | Sub-Saharan Africa (SSA) |
| ROW | middle income | Middle East and North Africa (MEA), reforming countries (REF) |

Definition of aggregated regions used in this study, income group, and the corresponding MAgPIE regions (see Fig. S2 for a map of MAgPIE regions). OECD90 refers to OECD countries as of 1990. ROW is an abbreviation for the rest of the world. Acronyms are explained the methods section.

**Table 3 | Grouping of scenario assumptions**

| Socio-economic drivers and dietary change | | Population | Income | Diet | Food Waste (% of food supply) | Timber Demand |
|---|---|---|---|---|---|---|
| SustDemand | off | SSP4 | SSP4 | SSP4 | no constraint | SSP4 |
| | on | SSP1 | SSP1 | EAT-Lancet | limited to 20% | SSP1 |
| **Environmental protection** | | **Land protection** | **Cropland set-aside** | **Envir. flow protection** | **Animal waste management** | **Fertilizer efficiency** |
| EnvirProt | off | WDPA | 0% | off | default | default |
| | on | WDPA+FF+BH | 20% | on | improved | improved |
| **Land-based mitigation** | | **AFOLU GHG price** | **Afforestation NDC** | **Afforestation $CO_2$ price** | **Afforestation limit** | **Bioenergy** |
| GHG Price | off | off | on | off | - | on |
| | on | on | on | on | 500 Mha globally | on |

Scenario assumptions are grouped into socio-economic drivers and dietary change (SustDemand on/off), environmental protection (EnvirProt on/off), and land-based mitigation (GHG Price on/off).

In the scenario *Global-Inequality*, global net AFOLU GHG emissions (SDG 13) remain positive until 2100, with levels of 8 Gt $CO_2$ eq $yr^{-1}$ in 2050 and 5 Gt $CO_2$ eq $yr^{-1}$ in 2100, which is considerably higher than 1.5 °C compatible projections of AFOLU GHG emissions (Fig. 6). Emissions in *Global-Inequality* are largely caused by non-$CO_2$ emissions from agriculture in Asia, Latin America, and Sub-Saharan Africa, with global levels of 2.4 Gt $CO_2$ eq $yr^{-1}$ for $N_2O$ and 5.5 Gt $CO_2$ eq $yr^{-1}$ for $CH_4$ in 2050 (Fig. 3). Global net $CO_2$ emissions from land-use change and management are around zero between 2040 and 2080, and slightly negative thereafter. However, this is the result of counteracting regional developments, where net $CO_2$ emissions from land-use change and management are (a) negative in high-income regions due to re/afforestation and (b) positive primarily in Sub-Saharan Africa due to continued expansion of cropland into primary and secondary forest (Fig. 4). Agricultural water use (SDG 6) remains close to or exceeds 2020 levels in all regions by 2050 (Fig. 8), with Asia accounting for ~58% of global agricultural water use in 2050 (Fig. 3). Global nitrogen fixation (SDG 15) in 2050 amounts to 137 Mt N $yr^{-1}$, which is more than twice than the proposed planetary boundary of 62 Mt N $yr^{-1}$. About 65% of global nitrogen fixation comes from Asia and Latin America alone. In addition, the prevalence of underweight (SDG 2) remains at high levels of above 600 million people globally in the 21st century, mostly in Asia and Sub-Saharan Africa (Fig. 1). At the same time, the prevalence of obesity (SDG 3) increases between 2020 and 2100 from 842 to 1537 million people globally.

Global sustainable land-use practices (*Global-EnvirProt*) have comparatively small effects on AFOLU GHG emissions, resulting in a global trajectory that remains positive until 2100 with levels of 6.7 Gt $CO_2$ eq $yr^{-1}$ in 2050 and 3.1 Gt $CO_2$ eq $yr^{-1}$ in 2100 (Fig. 6). Thus, sustainable land-use practices alone are insufficient for AFOLU GHG mitigation in line with 1.5 °C pathways. However, there are co-benefits in terms of reduced deforestation, especially primary forest in Sub-Saharan Africa and Latin America, as well as nitrogen fixation, which is 23% lower at global level in 2050 compared to *Global-Inequality* (Fig. 3).

A global price on AFOLU GHG emissions (*Global-GHG Price*) shows considerable potential for reducing AFOLU GHG emissions. The global GHG emission trajectory turns net-negative from 2060 onwards (0.8 Gt $CO_2$ eq $yr^{-1}$ in 2050 and −2.2 Gt $CO_2$ eq $yr^{-1}$ in 2100), and thus can be considered in line with AFOLU GHG emission reductions required for 1.5 °C compatible pathways (Fig. 6). The global GHG price has three major effects on net $CO_2$ emissions from land-use change and management: (a) strong reduction of $CO_2$ emissions from land conversion in Sub-Saharan Africa, Latin America and Asia, (b) increased carbon sequestration in managed forests, including a shift of re/afforestation from OECD90 + EU to Latin America, and (c) reduced $CO_2$ emissions from drained peatlands through rewetting (Fig. 7). These processes result in net global carbon uptake of 4.2 Gt $CO_2$ eq $yr^{-1}$ in 2050 (Fig. 3). Technical mitigation options reduce global $N_2O$ and $CH_4$ emissions from agriculture by 2050 to 1.8 Gt $CO_2$ eq $yr^{-1}$ and 3.2 Gt $CO_2$ eq $yr^{-1}$,

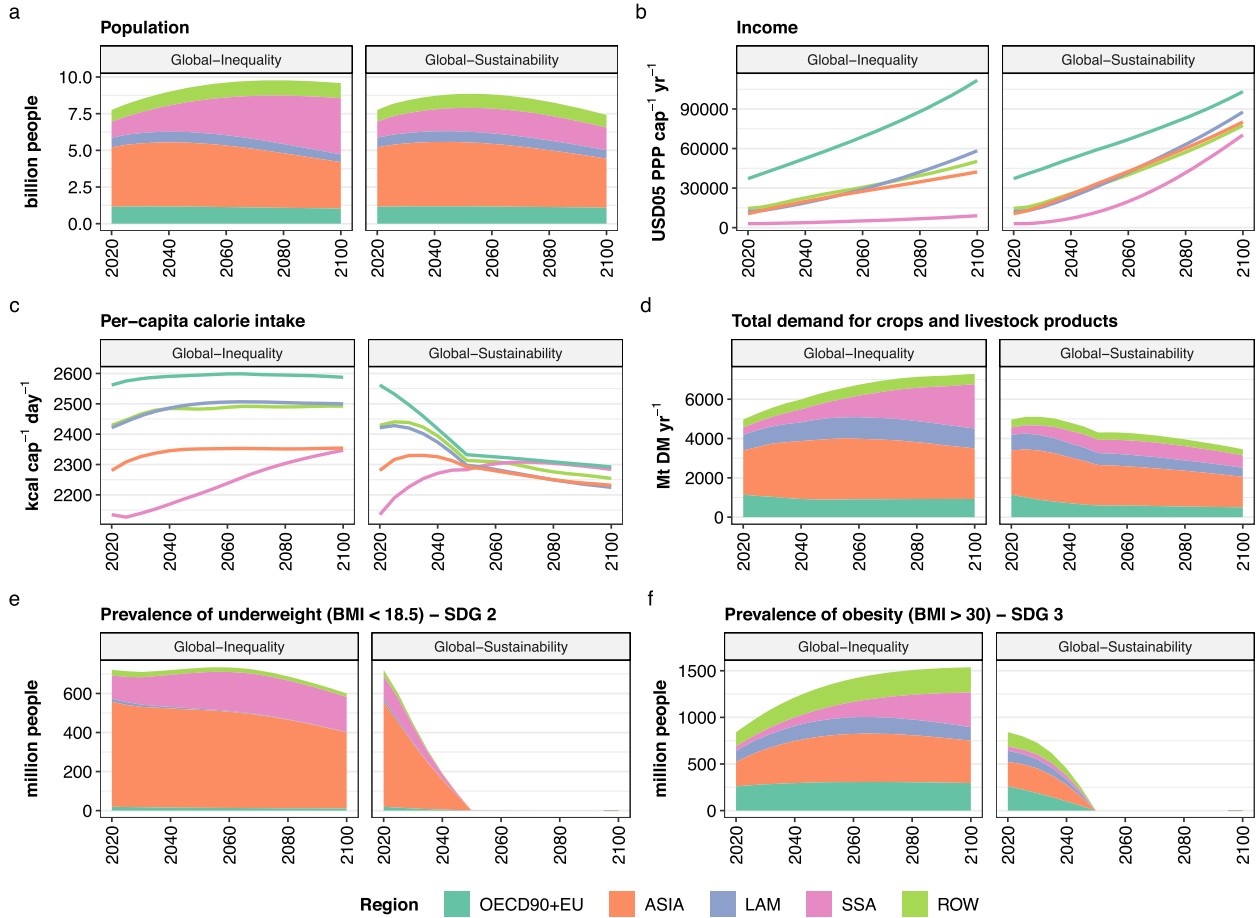

**Fig. 1 | Key socio-economic drivers and developments.** Data are shown at regional and global levels for the two main scenarios Global-Inequality (same for Global-GHG Price and Global-EnvirProt) and Global-Sustainability (same for Global-SustDemand). **a** population, **b** per-capita income, **c** per-capita calorie intake, **d** total demand for crops (including food and feed) and livestock products in million-ton dry matter per year, **e** prevalence of underweight (body mass index <18.5) and **f** prevalence of obesity (BMI > 30). Details on the prevalence of underweight and obesity can be found in Table S1.

respectively, which is 26% and 42% lower compared to *Global-Inequality*. However, AFOLU GHG emission pricing involves a trade-off with agricultural water use, which increases compared to 2020 levels and compared to *Global-Inequality* (Fig. 3). Agricultural water use increases especially in Asia, where environmental flows are violated in many places already today[23]. Moreover, long-term land-use intensification is higher in all regions compared to *Global-Inequality* (Fig. 5).

Global inclusive socio-economic development (*Global-SustDemand*) shows a similar global AFOLU GHG emission trajectory compared to *Global-GHG Price* (Fig. 6), and thus also can be considered as in line with 1.5 °C compatible AFOLU GHG mitigation. The combined effects of lower population growth, transition to healthy diets, and reduced food waste lower the demand for agricultural commodities (Fig. 1). In turn, the reduced pressure on limited land resources (a) strongly reduces $CO_2$ emissions from land conversion in Asia, Sub-Saharan Africa and Latin America, (b) increase carbon uptake from re/afforestation, and (c) reduces non-$CO_2$ emissions from agriculture in all regions (Figs. 3, 7). Global $N_2O$ and $CH_4$ emissions in 2050 amount to 1.2 Gt $CO_2$ eq yr$^{-1}$ and 2.2 Gt $CO_2$ eq yr$^{-1}$, respectively, which is about a 50% reduction compared to 2020 levels (Fig. 3). Net global carbon uptake amounts to 3.5 Gt $CO_2$ eq yr$^{-1}$ in 2050. Global inclusive socio-economic development shows co-benefits with several other SDGs beyond climate (SDG 13). Nitrogen fixation (SDG 15) and agricultural water use (SDG 6) are 26% and 22% lower, respectively, compared to *Global-Inequality* in 2050 at global level (Fig. 3). Moreover, the

prevalence of underweight (SDG 2), as well as the prevalence of obesity (SDG 3), are phased out by 2050 due to the assumed transition towards healthy diets (Fig. 1).

Global sustainable development (*Global-Sustainability*), which integrates sustainable land-use practices, AFOLU GHG emission pricing and inclusive socio-economic development in all regions, shows the strongest reduction of AFOLU GHG emissions (Fig. 6) and maximizes co-benefits with other SDGs (Figs. 1, 3). Net global carbon uptake amounts to 5.7 Gt $CO_2$ eq yr$^{-1}$ in 2050. Global $N_2O$ and $CH_4$ emissions decline to 0.7 Gt $CO_2$ eq yr$^{-1}$ and 0.9 Gt $CO_2$ eq yr$^{-1}$, respectively, by 2050. At the same time, nitrogen fixation (SDG 15) and agricultural water use (SDG 6) are 41% and 29% lower, respectively, compared to *Global-Inequality* in 2050 at global level. Identical to *Global-SustDemand*, the prevalence of underweight (SDG 2), as well as the prevalence of obesity (SDG 3), are phased out by 2050 due to the transition to healthy diets.

## Land-use change and intensification

Global and regional land-use change between 2020 and 2100 differs substantially between *Global-Inequality* and *Global-Sustainability* (Fig. 4a). In *Global-Inequality*, expansion of cropland and bioenergy area account for about half of total global land-use expansion by 2100, and timber plantations and afforestation for the other half. About half of total global land-use expansion relies on the reduction of pasture areas, which is facilitated by a shift from pasture-based towards

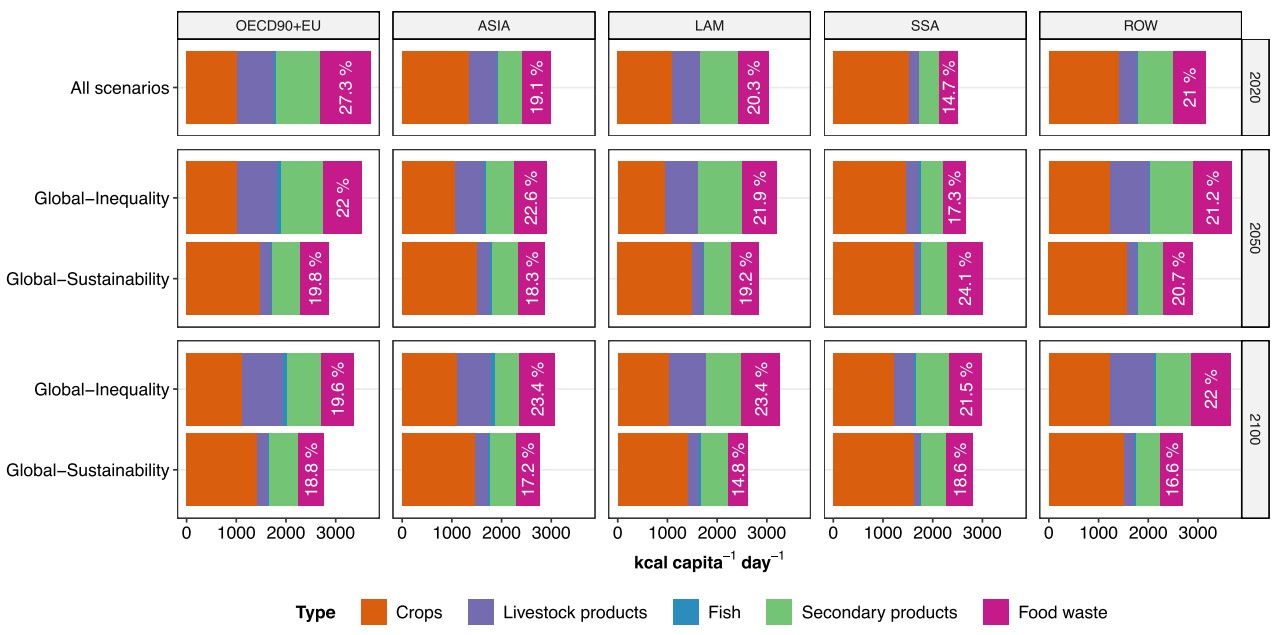

**Fig. 2 | Per-capita calorie supply.** The sum of crops, livestock products, fish, and secondary products reflects food intake, which together with food waste sums up to food supply. Percentage values indicate the share of food waste in the total food supply. Data are shown at the regional level for the two main scenarios Global-Inequality (same for Global-GHG Price and Global-EnvirProt) and Global-Sustainability (same for Global-SustDemand).

## Table 4 | Scenario definitions

| Scenario name | SustDemand | EnvirProt | | GHG Price | |
|---|---|---|---|---|---|
| | All world regions | OECD90 + EU | ASIA, LAM, SSA, ROW | OECD90 + EU | ASIA, LAM, SSA, ROW |
| Global-Inequality | off | on | off | on | off |
| Global-SustDemand | on | on | off | on | off |
| Global-EnvirProt | off | on | on | on | off |
| Global-GHG Price | off | on | off | on | on |
| Global-Sustainability | on | on | on | on | on |

Grouped assumptions (see Table 3) are turned on/off in the modeling for selected regions (see Table 2).

cropland-based animal feed in combination with increased pasture and livestock productivity over time. The other half relies on the conversion of potentially carbon-rich ecosystems such as primary and secondary forest, as well as non-forest natural land. These global developments are driven by heterogeneous regional dynamics (Fig. 4b). About 80% of global loss of primary forest, secondary forest, and non-forest natural land between 2020 and 2100 occur in Sub-Saharan Africa, while there is no loss in high-income regions, the only ones with a price on GHG emissions in *Global-Inequality*. In addition, the GHG price in high-income regions triggers endogenous afforestation with the goal of carbon dioxide removal (see methods). Prescribed afforestation according to nationally determined contributions (NDCs) is especially high in China. Timber plantations for wood production play an important role in high-income regions and Asia, while wood production in other regions largely relies on harvests from natural forests. Land-use intensification, an indicator of human-induced crop yield amplification due to technological change, increases in all regions over time, with higher growth rates in low- and middle-income regions compared to high-income regions (Fig. 5).

Global sustainable land-use practices in the scenario *Global-EnvirProt* strongly reduce the loss of primary forest in Sub-Saharan Africa and Latin America, which, however, comes at the cost of more conversion of secondary forest. Besides this difference, overall land-use dynamics as well as land-use intensification in *Global-EnvirProt* are comparable to *Global-Inequality* (Figs. 4, 5).

A globally coordinated price on AFOLU GHG emissions in the scenario *Global-GHG Price* strongly reduces deforestation and conversion of non-forest natural land in all regions. At the same time, the global GHG price shifts re/afforestation from Europe and Northern America to Latin America, which has a higher potential for carbon sequestration. To compensate for reduced land conversion, there is a shift away from cropland expansion towards higher land-use intensification for supplying the same food, feed, bioenergy, and timber as in *Global-Inequality* (Figs. 4, 5).

Global inclusive socio-economic development in the *Global-SustDemand* scenario strongly reduces deforestation and conversion of non-forest natural land in all regions, comparable to *Global-GHG Price*. The declining crop and livestock demand (Fig. 1) also frees-up land for more re/afforestation in high-income regions (the only ones with a GHG price in this scenario). At the same time, land-use intensification is lower compared to *Global-Inequality* due to the reduced pressure in the land-use system (Figs. 4, 5).

In the combined scenario, *Global-Sustainability*, declining food demand, and global GHG emission pricing interact, which virtually brings deforestation to halt in all regions, increases non-forest natural land in all regions, and increases re/afforestation mostly in Latin America. At the same time, land-use intensification is comparable to the relatively low rates of *Global-SustDemand* (i.e., lower than in *Global-Inequality*) in all regions except Asia, where the land-limiting

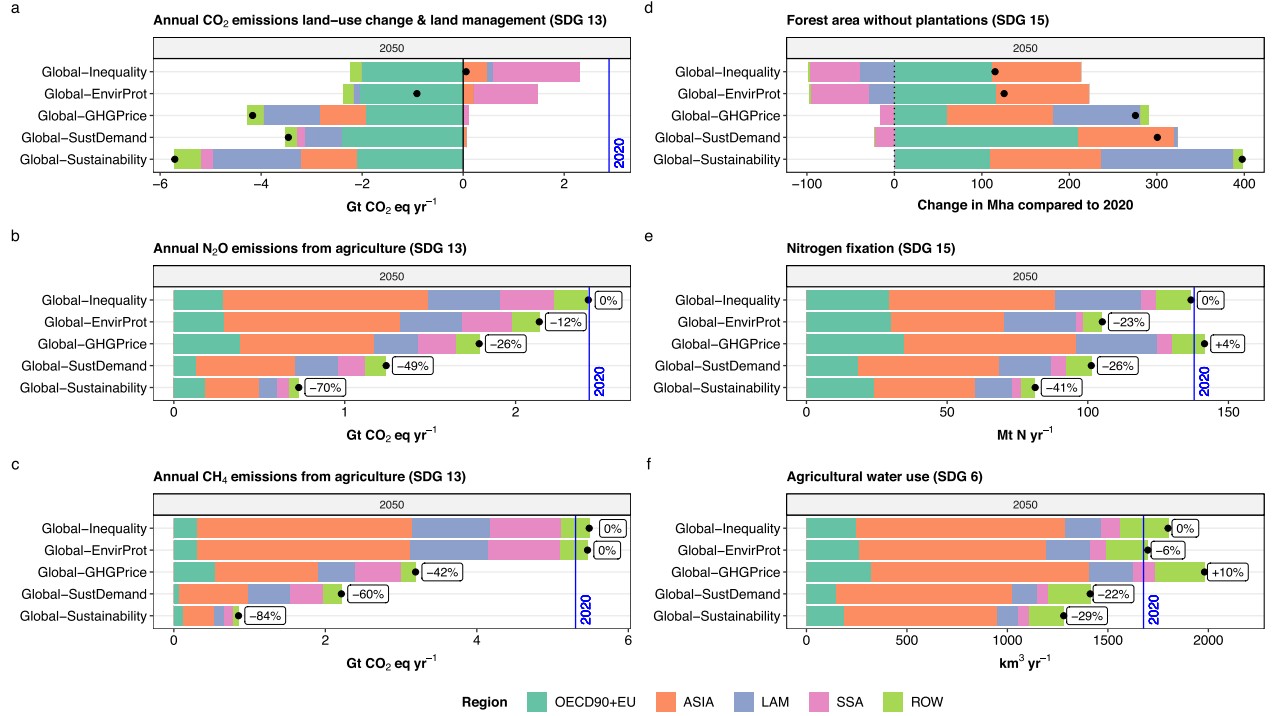

**Fig. 3 | Summary of results for five scenarios and six indicators (mapped to SDGs) for 2050 at regional and global level. a** $CO_2$ emissions and removals from land-use change and management, **b** $N_2O$ emissions from agriculture, **c** $CH_4$ emissions from agriculture, **d** change in forest area without plantations, **e** nitrogen fixation, and **f** agricultural water use. Black dots show the global net effect for each indicator and scenario. Blue vertical lines show the respective value in year 2020.

Percentage labels in **b**, **c**, **e** and **f** show the relative change of the global indicator level for each scenario compared to the Global-Inequality scenario. For **a** and **d**, it is not meaningful to show percentage changes because these indicators can, by definition, turn net-zero (in case of divergent regional dynamics) or net-negative at global level.

effect of GHG emission pricing results in higher rates of land-use intensification (Figs. 4b, 5).

## AFOLU GHG emissions

Net annual $CO_2$ emissions from land-use change and management as well as $N_2O$ and $CH_4$ emissions from agriculture differ considerably between *Global-Inequality* and *Global-Sustainability* (Fig. 6a). Global net AFOLU GHG emissions remain positive in *Global-Inequality* throughout the 21st century with levels of 8 Gt $CO_2$ eq yr⁻¹ in 2050 and 5 Gt $CO_2$ eq yr⁻¹ in 2100. These GHG emissions are largely caused by $N_2O$ and $CH_4$ emissions from agriculture, while global net $CO_2$ emissions from land-use change and management are close to zero (Fig. 7). $N_2O$ and $CH_4$ emissions decrease in high-income regions but continue or even increase in low- and middle-income regions until 2050. By 2100, $N_2O$ and $CH_4$ emissions decrease in all regions except Sub-Saharan Africa, mostly due to population dynamics in combination with per-capita livestock product consumption patterns (Fig. 1). Global net $CO_2$ emissions from land-use change and management are close to zero mostly because of re/afforestation-based carbon sequestration in high-income regions that compensates for increasing $CO_2$ emissions from deforestation in Sub-Saharan Africa (Fig. 6b). In contrast, global net AFOLU GHG emissions turn net-negative in *Global-Sustainability* from 2035 onwards, reaching levels of −4.1 Gt $CO_2$ eq yr⁻¹ in 2050 and −6.9 Gt $CO_2$ eq yr⁻¹ in 2100 (Fig. 6a). This development is largely due to declining $N_2O$ and $CH_4$ emissions in all world regions in combination with net-negative $CO_2$ emissions from effective forest protection and economic incentives for re/afforestation.

The reduction of AFOLU GHG emissions in *Global-Sustainability* is facilitated by two major factors, one on the demand side (*Global-SustDemand*) and one on the supply side (*Global-GHG Price*). On the demand side, lower population growth, a transition to healthy diets

and reduced food waste lower the demand for crops and livestock products (Fig. 1). In turn, these developments (a) strongly reduce $CO_2$ emissions from land conversion and wood harvest, (b) increase carbon uptake from re/afforestation, and (c) reduce non-$CO_2$ emissions from agriculture, including $N_2O$ emissions from agricultural soils (fertilizer application) and animal waste management as well as $CH_4$ emissions from enteric fermentation, animal waste management and rice cultivation (Fig. 7). On the supply side, the global GHG price (a) strongly reduces the conversion of carbon-rich forests and other ecosystems, (b) increases carbon sequestration in managed forests (re/afforestation and timber plantations), and (c) reduces $CO_2$ emissions from managed peatlands trough rewetting (Fig. 7). Moreover, the GHG price activates the following technical mitigation options for non-$CO_2$ emissions: (1) changes in animal feed for reducing $CH_4$ emissions from enteric fermentation, (2) anaerobic digesters for reducing $CH_4$ and $N_2O$ emissions from animal waste management, (3) improved water management for reducing $CH_4$ emissions from rice cultivation and (4) improved fertilizer application for reducing $N_2O$ emissions from agricultural soils. Both options, AFOLU GHG emission pricing as well as inclusive socio-economic development, show considerable potential for AFOLU GHG emission reduction by 2050 and beyond, especially in Asia, Latin America, and Sub-Saharan Africa, resulting in net-negative global AFOLU GHG emissions from around 2050 onwards (Fig. 6). In contrast, global sustainable land-use practices (*Global-EnvirProt*) have comparatively small effects on AFOLU GHG emissions, resulting in a global emission trajectory that remains positive and relatively close to *Global-Inequality* until 2100. For comparison, in 1.5 °C compatible transformation pathways global net AFOLU GHG emissions decline to 2.6 Gt $CO_2$ eq yr⁻¹ by 2050 and −1.1 Gt $CO_2$ eq yr⁻¹ by 2100 (median values across three illustrative mitigation pathways from the IPCC AR6 Scenarios Database: LD, Ren, and SP)[24].

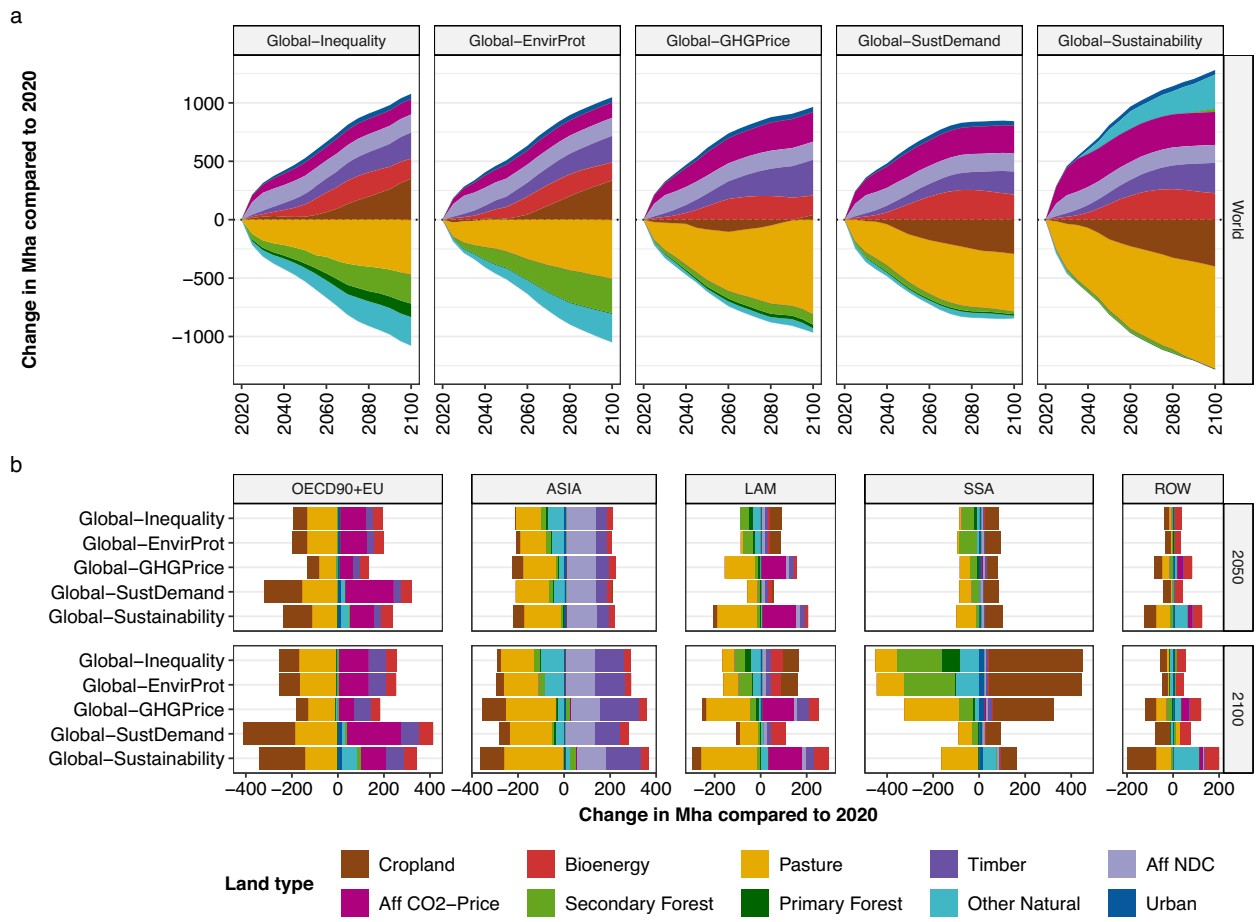

**Fig. 4 | Land-use change of major land types over time compared to 2020 for five scenarios. a** show results at global level over time. **b** shows results at regional level for 2050 and 2100. Cropland includes food, non-food, and feed crops. Bioenergy includes 2nd generation bioenergy (fast growing grasses and trees such as miscanthus and poplar). Pasture includes rangeland and managed pasture areas. Timber refers to timber plantations for wood production. Aff NDC includes prescribed re/afforestation according to National Determined Contributions (NDCs) towards the objectives of the Paris Agreement. Aff CO2-Price refers to endogenous re/afforestation-based on the underlying scenario-specific $CO_2$ price trajectory. Secondary forest includes modified and regrown forest, e.g., following wood harvest or cropland abandonment. Primary forest is intact forest without signs of human intervention. Urban land includes built-up area. Other natural land is a residual category, which includes among others non-forest ecosystems, deserts, and shrublands.

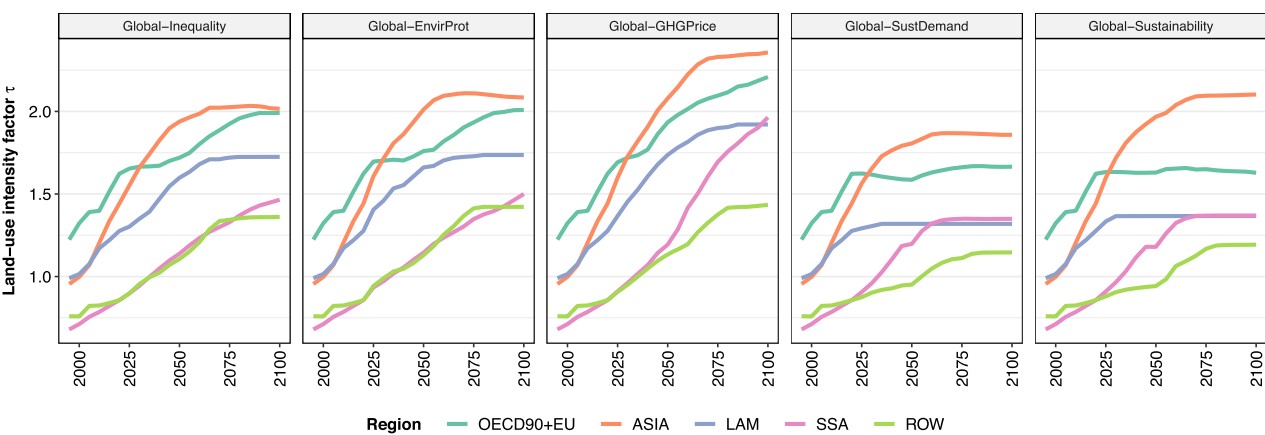

**Fig. 5 | Land-use intensity factor τ.** Data are shown at regional level for five scenarios. The τ factor reflects the degree of crop yield amplification caused by human activities. A duplication of τ implies a doubling of crop yields under fixed environmental conditions. See Fig. S10 for validation data.

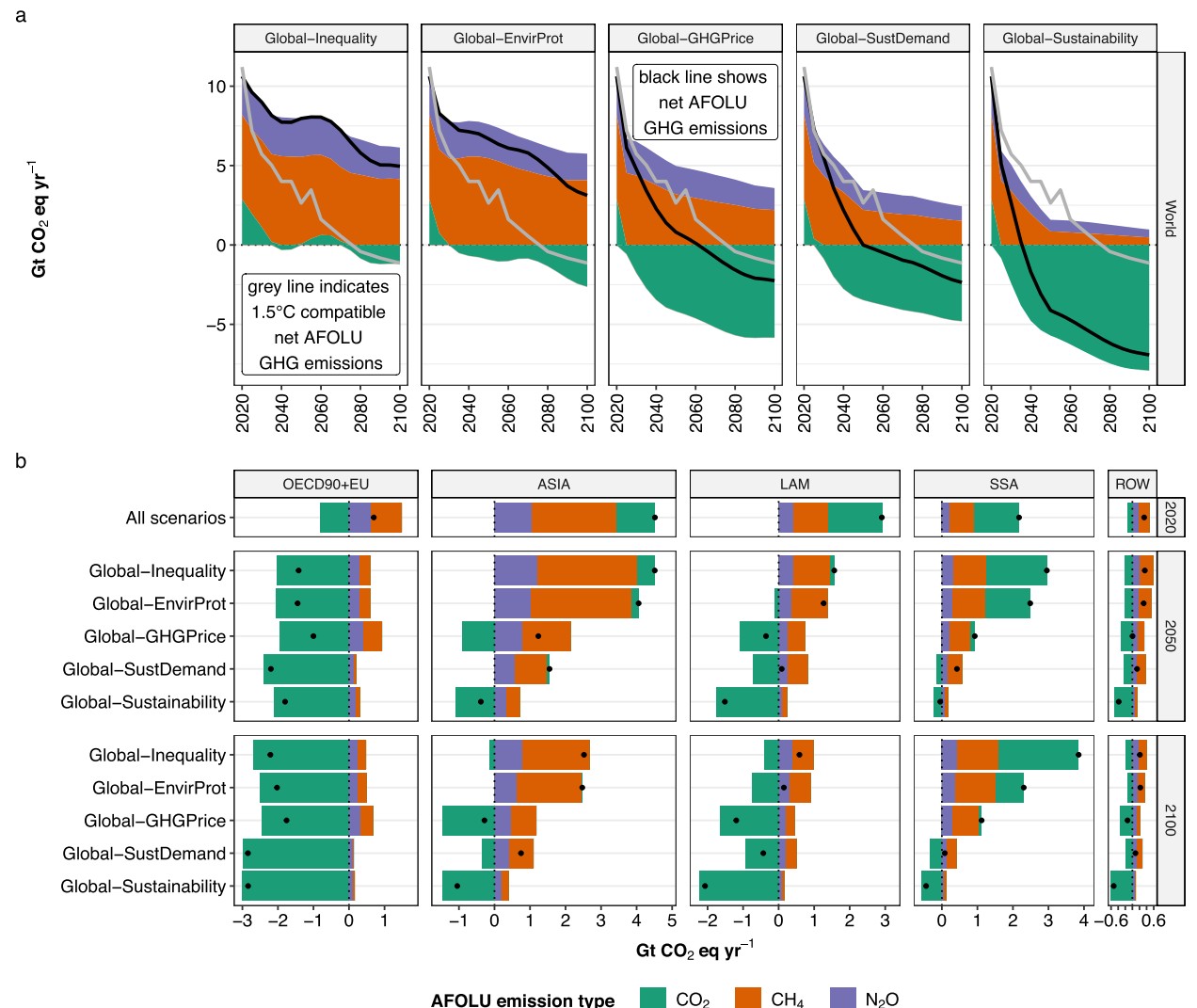

**Fig. 6 | AFOLU GHG emissions over time for five scenarios. a** shows results at global level over time. **b** shows results at regional level for 2020, 2050, and 2100. Black solid lines in **a** and dots in **b** show the net effect across AFOLU GHG emissions. Gray solid lines in **a** show median values across 1.5 °C compatible illustrative mitigation pathways (LD, Ren, SP) from the IPCC AR6 Scenarios Database[24]. $CO_2$ includes emissions from land-use change such as deforestation and conversion of non-forest ecosystems, as well as removals from re/afforestation and natural succession on abandoned agricultural land. $CH_4$ includes emissions from enteric fermentation, animal waste management, and rice cultivation. $N_2O$ includes emissions from agricultural soils (fertilizer application) and animal waste management. $N_2O$ and $CH_4$ emissions have been converted into $CO_2$ equivalents using IPCC AR6 GWP100 factors of 273 and 27, respectively.

## Water use, nitrogen losses, and forest area

Today, agriculture accounts on average for 70% of all freshwater withdrawals globally, and in many parts of the world, especially in Asia, water withdrawals already tap into environmental flow requirements (SDG 6)[23]. In the *Global-Inequality* scenario, agricultural water use remains close to or exceeds 2020 levels in all regions by 2050 (Fig. 8). By 2100, agricultural water use increases especially in high-income regions but also in Latin America and Sub-Saharan Africa, while Asia shows a reduction. In *Global-Sustainability*, in contrast, future agricultural water use is lower compared to 2020 levels in all regions except Sub-Saharan Africa where agricultural water use remains close to 2020 levels. Asia shows by far the strongest reduction, followed by high-income regions. At the global level, agricultural water use in 2050 is 29% lower in *Global-Sustainability* compared to *Global-Inequality* (Fig. 3f). This reduction is largely facilitated by inclusive socio-economic development (*Global-SustDemand*). While the protection of environmental flows in *Global-EnvirProt* also reduces agricultural water use in Asia to some extent, it increases water use in Latin America

and Sub-Saharan Africa in the long term (Fig. 8). This is because *Global-EnvirProt* also includes the protection of frontier forests and biodiversity hotspots, which limits the expansion of agricultural land into high-yielding forest areas, which in turn leads to the intensification of production on existing cropland via irrigation. A global GHG price (*Global-GHG Price*), which also limits the conversion of high-yielding forest areas, considerably amplifies this effect of increased agricultural water use in Latin America and Sub-Saharan Africa, besides higher rates of yield-increasing technological change (Fig. 5).

Nitrogen fixation is a proxy for nitrogen losses to the environment and hence ecosystem degradation (SDG 15). In our scenarios, global nitrogen fixation amounts to 138 Mt N yr$^{-1}$ in 2020, which is more than twice as high as a proposed global planetary boundary of 62 Mt N yr$^{-1}$ [25,26]. In *Global-Inequality*, future nitrogen fixation decreases compared to 2020 levels in high-income regions and Asia but increases in Latin America and Sub-Saharan Africa (Fig. 8). These regional developments result in a global value of 137 Mt N yr$^{-1}$ for 2050, and thus almost unchanged

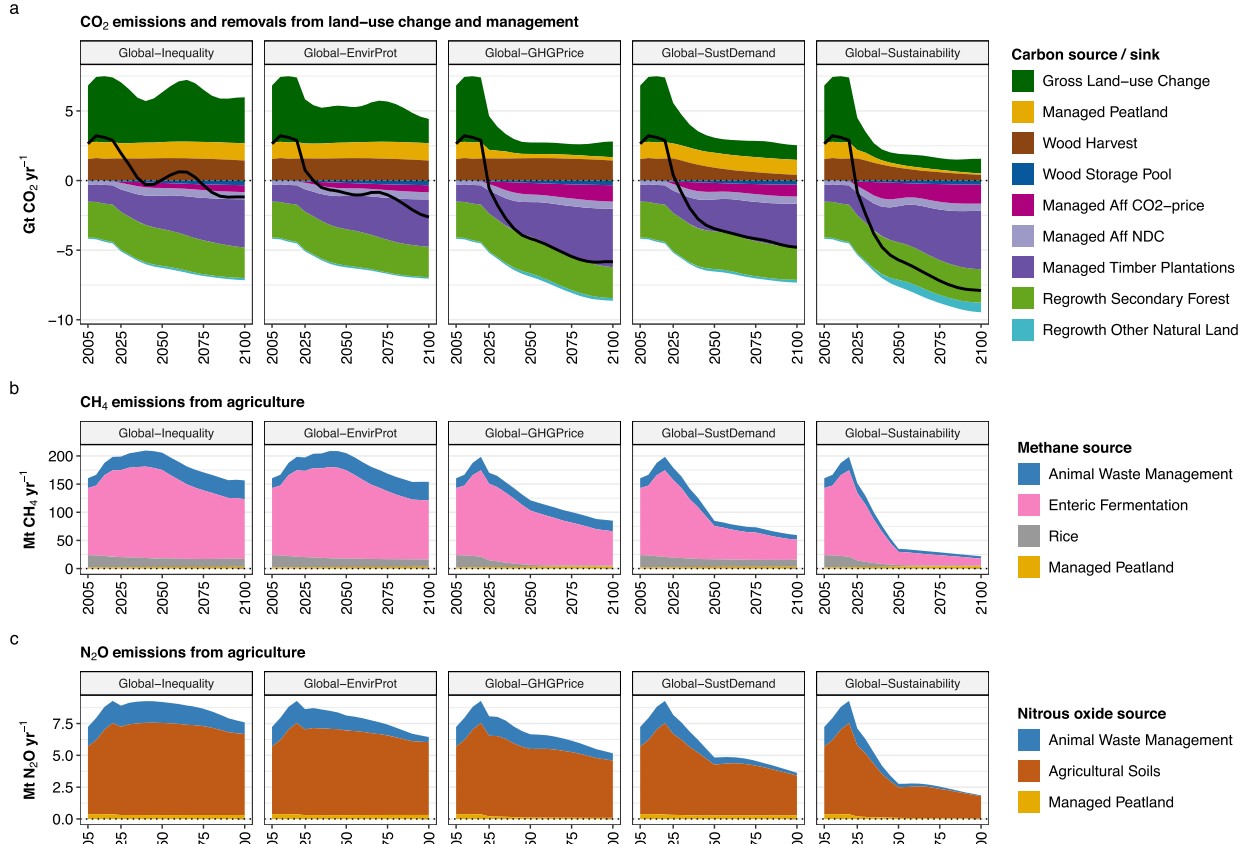

**Fig. 7 | AFOLU GHG emission breakdown by source.** Data are shown at global level for five scenarios. **a** shows $CO_2$ emissions and removals from land-use change and management. Carbon losses consist of emissions from deforestation, conversion of non-forest ecosystems, drained peatlands, and wood harvest. Carbon gains consist of carbon storage in wood products, re/afforestation (Aff CO2-price and Aff NDC), timber plantations, and regrowth of natural vegetation (secondary forests and other natural land). The black line shows the net effect of carbon losses and carbon gains at the global level. **b** shows $CH_4$ emissions from agriculture. **c** shows $N_2O$ emissions from agriculture. Further details on AFOLU GHG emission sources and sinks are provided in the methods section and in Table S1.

compared to 2020. In *Global-Sustainability*, nitrogen fixation is lower compared to 2020 levels in all regions, with the strongest reductions in high-income regions and Asia. At the global level, nitrogen fixation declines to 81 Mt N yr$^{-1}$ by 2050, which reduces the gap compared to the planetary boundary value (Fig. 3e). The reduction of nitrogen fixation in *Global-Sustainability* is facilitated by inclusive socio-economic development (*Global-SustDemand*) and sustainable land-use practices (*Global-EnvirProt*), which reduce nitrogen fixation by 26% and 23%, respectively (Fig. 3).

Net change in forest area (excluding timber plantations) is an indicator for the conservation and restoration of terrestrial ecosystems (SGD 15). Deforestation is reduced to zero in all regions in *Global-Sustainability* by 2050, whereas in *Global-Inequality* Latin America and especially Sub-Saharan Africa exhibit considerable loss of primary and secondary forests (Figs. 4, 8). In addition, Latin America shows net forest gains in *Global-Sustainability*. Net forest cover in high-income regions and Asia is expanding in both scenarios due to existing national polices, and in the case of high-income regions also due to the AFOLU GHG price. At the global level, net forest cover increases between 2020 and 2050 by 398 Mha in *Global-Sustainability* and 115 Mha in *Global-Inequality* (Fig. 3d). The higher increase of forest cover in *Global-Sustainability* is facilitated by two major factors: (a) a global GHG price (*Global-GHG Price*), which provides an economic incentive for forest protection and restoration, and (b) inclusive socio-economic development (*Global-SustDemand*), which lowers total crop

and livestock demand and hence frees-up land resources for re/afforestation. While the outcome of both scenarios is similar at the global level, the regional allocation of re/afforestation differs considerably (Figs. 3, 8). Re/afforestation area roughly doubles in OECD90 + EU under inclusive socio-economic development compared to *Global inequality* (Fig. 4) due to the combined effects of GHG emission pricing (OECD90 + EU has a GHG price in all scenarios) and reduced pressure on land from food production. In contrast, a global GHG price shift re/afforestation from Europe and Northern America to Latin America, which has a higher potential for terrestrial carbon sequestration.

## Discussion
In this study, we compare a scenario of global inequality, where AFOLU GHG emission pricing and sustainable land-use practices remain limited to high-income regions, to a scenario of global sustainability with AFOLU GHG emission pricing, sustainable land-use practices as well as inclusive socio-economic development in all world regions. Our results indicate that in a world of global inequality, future land-use trajectories and agricultural production in low- and middle-income regions could cause considerable GHG emissions, which would prevent global net AFOLU GHG emissions from declining towards 1.5 °C compatible levels by 2050 and 2100. The decomposition of scenario drivers indicates that AFOLU GHG emission pricing as well as inclusive socio-economic development could reduce global net AFOLU GHG emissions to 1.5 °C compatible levels. However, only inclusive socio-

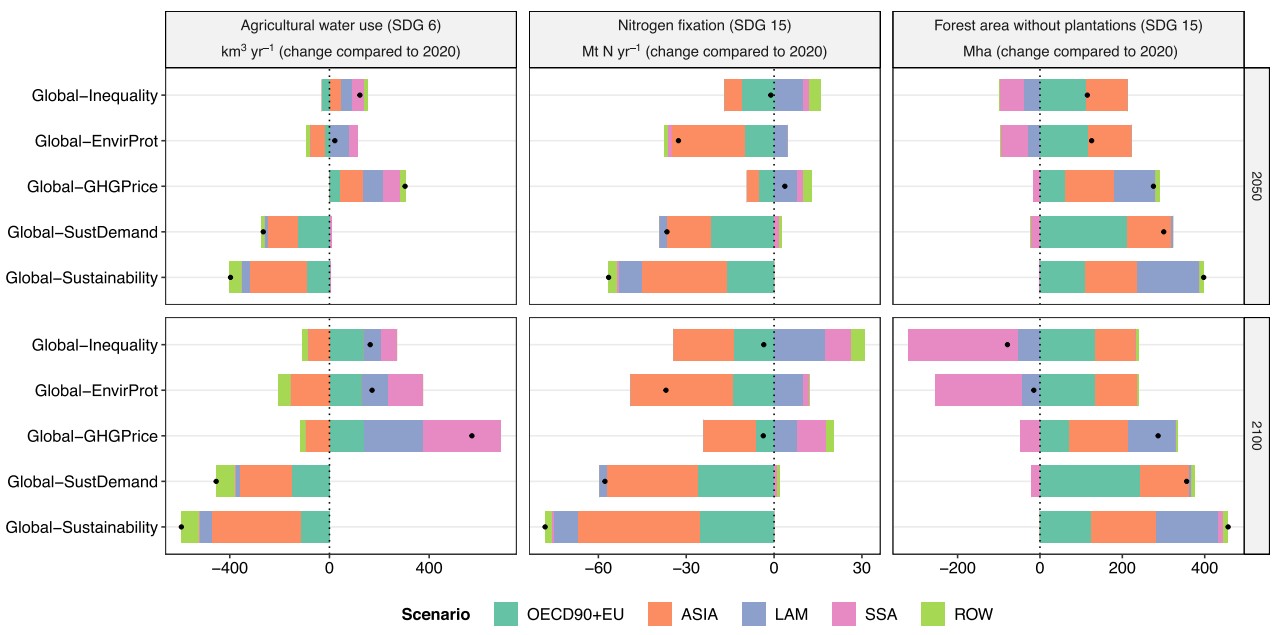

**Fig. 8 | Change in agricultural water use, nitrogen fixation, and forest area.** Data are shown relative to 2020 for five scenarios at regional level for the years 2050 and 2100. The black dot indicates the net effect at the global level.

economic development comes with multiple co-benefits, especially in low- and middle-income regions, for other SDGs than climate protection (SDG 13), including malnutrition (SDG 2), overnutrition (SDG 4), ecosystem conservation and restoration (SDG 15), nitrogen pollution (SDG 15) and agricultural water use (SDG 6). The sole implementation of sustainable land use practices globally would overall have substantially more limited benefits for global AFOLU GHG emissions and the investigated SDG indicators.

Our study is deliberately focused on the land sector to inform the debate on AFOLU GHG mitigation. This implies that we do not account for interactions of AFOLU with other sectors such as energy and transport, which is especially relevant for biomass supply[27]. AFOLU GHG emissions in mitigation pathways strongly depend on scenario assumptions in terms of population, income, diets, environmental protection, and land-use regulation[13]. As a benchmark for 1.5 °C compatible global AFOLU GHG emissions, we use median values across three illustrative mitigation pathways from the IPCC AR6 Scenarios Database (LD, Ren, SP)[24]. Under a given carbon budget or temperature target for 1.5 °C, AFOLU GHG emissions above the benchmark like in our *Global-Inequality* scenario might be balanced by more mitigation in other sectors. However, considering that total GHG emissions need to decline rapidly for compliance with 1.5 °C pathways, shifting mitigation efforts from AFOLU to other sectors is only possible to a limited extent[11]. Especially carbon dioxide removal in the AFOLU sector via re/afforestation plays a central role in the cost-effective balancing of 'hard-to-abate' residual emissions (e.g., from agriculture, shipping, and aviation) in 1.5 °C pathways[11]. On the contrary, lower AFOLU GHG emissions like in our *Global-Sustainability* scenario could reduce the mitigation burden of other sectors.

A core assumption in our scenarios with global inclusive socio-economic development is that there will be progress towards achieving SDGs 2–4 on zero hunger, health, and education (transition to healthy diets, reduced food waste, and lower population growth). Economic development alone might be insufficient to achieve the transition required, with a strengthening of governance and institutions (SDG 16) and inclusive, gender-equal development (SDG 4 and 5) being key enabling factors[28,29]. Estimates of the investment gap (i.e., the additional investments required above current trends) for meeting

SDGs 2–4 in low- and middle-income countries are in the range of up to a few hundred billion USD per year for each of these SDGs[30]. In addition, the investment gap for SDG 13 on climate action is estimated to range from 0.3 to 3 trillion USD per year globally[30]. Future research could explore the potential of development finance, international climate policy, and climate partnerships for reducing these large investment gaps. A complementary option to reduce investment gaps for inclusive socio-economic development is domestic carbon pricing, which not only reduces emissions but also generates revenues[10]. For instance, the projected revenues of carbon pricing would be sufficient to finance more than two-thirds of the entire public SDG financing needs in several countries in South- and Southeast Asia, while the potential is substantially lower in most African countries[10]. Thus, inclusive socio-economic development and GHG emission pricing in the AFOLU sector, which we identified as the two most effective options for reducing AFOLU GHG emissions in low- and middle-income countries, might be mutually reinforcing also from a financial perspective. Our results highlight that identifying and rapidly implementing such ways of synergically addressing climate action and inclusive sustainable development globally is critical for the achievement of land-based mitigation compatible with the Paris Agreement objectives.

## Methods
### Land-use model MAgPIE
In this study, we use the global multi-regional MAgPIE 4 open-source land-use modeling framework for generating scenarios[20]. The model version we use here is MAgPIE 4.4.0[21] (see code availability statement). MAgPIE combines economic and biophysical approaches to simulate spatially explicit global scenarios of land use within the 21st century and the respective interactions with the environment (Fig. S1). The MAgPIE framework has been used to simulate mitigation pathways for different Shared Socio-economic Pathways (SSPs)[13] and contributed to several IPCC reports[1,31,32].

MAgPIE is a global partial equilibrium model of the land-use sector, covering agriculture and forestry, that operates in a recursive dynamic mode and incorporates spatially explicit information on biophysical constraints into an economic decision-making process. It

takes regional economic conditions such as demand for agricultural commodities, technological development, and production costs as well as spatially explicit data on biophysical constraints into account. Geographically explicit data on biophysical conditions are provided by the Lund-Potsdam-Jena managed land model (LPJmL)[33–36] on a 0.5-degree resolution and include e.g., carbon densities of different vegetation types, agricultural productivity such as crop yields, and water availability for irrigation. Due to computational constraints, all model inputs in 0.5 degree resolution are aggregated to simulation units for the optimization process based on a clustering algorithm[37]. In this study, we apply a novel two-staged approach. In the first step, the model is solved globally with 200 simulations units (clusters), which allows capturing trade patterns between 12 economic regions. In the second step, we use the trade balance from the first step as exogenous input and solve the 12 economic regions individually (parallel optimization), which allows to use 2000 simulations units in total, thus improving the native spatial resolution in MAgPIE by a factor of 10 (Fig. S2). Available land types in MAgPIE include cropland (food, feed, material, and bioenergy), pasture, timber plantations, forest (primary and secondary), other land (non-forest vegetation, abandoned agricultural land, and deserts), and urban land. Cropland, pasture, timber plantations, forest, and other land are endogenously determined, while the development of urban areas follows exogenous SSP-based assumptions. Crop yield increases due to technological change are modeled endogenously based on regionally different investment-yield ratios and interest rates[38]. Hence, the model simultaneously optimizes the rate of yield-increasing technological change and cropland expansion, which is especially relevant for long-term projections. As indicator of yield-increasing technological change, we calculate the output-related land-use intensity factor $\tau$, which reflects the degree of crop yield amplification caused by human activities[39]. Cropland covers the cultivation of different crop types (e.g., temperate and tropical cereals, maize, rice, oilseeds), both rainfed and irrigated systems, and two second-generation bioenergy crop types (grassy and woody). International trade is based on historical trade patterns and economic competitiveness. Food demand is derived based on population growth and dietary transitions, accounting for changes in food waste and intake, with shifting shares of animal calories, processed products, fruits, and vegetables as well as staples. Timber demand is based on population and income dynamics[40]. For the analysis in this study, we aggregate the 12 native model regions of MAgPIE into 5 broader regions (Table 2, Fig. S2).

Annual net $CO_2$ emissions from land-use change and management are calculated based on changes in carbon stocks of vegetation, and therefore may vary substantially between time steps (stock-flow problem). To avoid our results being biased by the values of single years, we calculate in a post-processing step an average value by applying a low-pass filter that distributes annual net $CO_2$ emissions over time, while making sure that the time integral over the modeled period remains the same. Carbon stock changes in vegetation are subject to land-use change dynamics such as the conversion of forest into agricultural land[41]. In case of re/afforestation or when agricultural land is taken out of production, regrowth of natural vegetation removes $CO_2$ from the atmosphere. $N_2O$ emissions from agricultural soils (fertilizer application) and animal waste management are estimated based on nitrogen budgets for croplands, pastures, and the livestock sector[42,43]. $CH_4$ emissions from agriculture include emissions from enteric fermentation, animal waste management, and rice cultivation, which are estimated based on feed demand, manure, and rice cultivation area, respectively[42,44]. $CO_2$, $CH_4$, and $N_2O$ emissions from managed peatlands (drained and rewetted) are calculated based on IPCC wetland GHG emission factors[45]. The calculation of $CO_2$ emissions from wood harvest, carbon storage in wood products, and carbon uptake in timber plantations follows the methodology described in Mishra et al.[40]. In mitigation scenarios, AFOLU GHG emissions are subject to pricing.

$CO_2$ emissions are reduced endogenously through reduced conversion of forests, other natural lands, and intact peatlands, while $CH_4$ and $N_2O$ emissions are reduced based on marginal abatement cost curves[44–46]. In addition, the $CO_2$ price serves as an incentive for re/afforestation and peatland restoration (rewetting of degraded peatlands)[45,47]. Validation of main land types and GHG emissions is available in the SI (Figs. S7, S8).

## Scenario co-development process

The narratives of the scenarios investigated in this study have been co-developed as part of the LAMACLIMA (short for LAnd MAnagement for CLImate Mitigation and Adaptation) project, which aims to advance the scientific and public understanding of the interactions between climate and changes in land cover and management, and to help elaborate sustainable land-based adaptation and mitigation measures. On the one hand, the co-development process involved scientists from the project consortium with diverse profiles: Earth system modelers whose expertize lie in land-atmosphere interactions, land-use modelers, economists specialized in climate impacts, and stakeholder engagement experts. They worked together with a group of stakeholders constituted of land-use experts with diverse backgrounds: academia (with expertize in various fields of land-use and climate science), environmental consultancies, international organizations, European agencies, and governmental bodies. Most stakeholders were based in European or North American countries, with the focus of their activities mostly also lying in those regions.

The scenario co-development process started with three online meetings of 1.5 hours each held in February and March 2021, which focused on introducing to the stakeholders the main research questions tackled by the LAMACLIMA project, the tools used to investigate them, as well as the first results. Then, the bulk of the co-development occurred during a 2-day online workshop of twice 5-hour sessions organized on April 14-15 2021.

The first objective of this process was to identify scenario narratives considered of interest and broad policy relevance to the stakeholders, which reflect different ways to meet climate and environmental policy objectives and would result in different evolutions of land use across the world. Moreover, these narratives were meant to feed into other activities of the LAMACLIMA project, namely the modeling of the resulting land-use trajectories with MAgPIE, and in turn the climate impacts thereof using Earth system models (ESMs). Therefore, a second objective of the co-development process was to identify how the narratives could be implemented in the MAgPIE model while retaining the characteristics of their distinct essences.

The pursuit of this dual objective led to an iterative co-development process during which many cycles of interactions took place between stakeholders and project scientists, which were grossly structured as follows. Information was presented to the stakeholders about the processes that the models used in the LAMACLIMA project can represent and the types of questions that they can help answer, followed by a brainstorming session during which the stakeholders' listed aspects that they found of potential interest and policy relevance and worth considering in the scenario narratives. This helped both groups identify which of these aspects could be represented in MAgPIE and ESMs (or not). This in turn led the participants to this process to revisit the original stakeholders' list of interests, narrowing it down but also potentially triggering additional ideas, leading to another such cycle. The reoccurrence of such iterations along the online meetings and workshop reflects the efforts required for the project scientists and the more diverse group of stakeholders to get to know the objectives and boundary conditions of the co-development process as well as each other's interests and domain of expertize. Overall, the resulting scenario narratives reflect the necessity to find a common ground between a long list of aspects of potential policy relevance and interest to the stakeholders, the more limited possibility to represent

these aspects in the models used in the project, and the desire to derive scenario narratives for which the corresponding land-use trajectories (simulated by MAgPIE and described in this study) and impacts on both the carbon cycle and the climate via biogeophysical effects (to be in turn simulated by ESMs) exhibit enough differences to be able to draw robust scientific conclusions. However, despite the mentioned limited possibilities of the models employed in the LAMACLIMA project, an important co-benefit of the co-development process was the resulting two-way capacity building it enabled to happen: while stakeholders were able to learn about new scientific results and the functioning of state-of-the-art models used in the project, the project scientists were exposed to scientific questions and aspects of interest to stakeholders, thereby also triggering ideas to guide future research.

Given that the activities of most participating stakeholders lie in the EU or other countries with advanced economies, and given the world's division into 12 regions in MAgPIE, the co-development process resulted in the decision to develop scenarios that both achieve climate and environmental objectives in the European union (EU) as well as the Organization for Economic Co-operation and Development (OECD) countries (referred to as OECD90 + EU or high-income regions). Although the stakeholders highlighted that these objectives could be reached via very different land-use strategies ('land sparing' and 'land sharing' were for example explicitly mentioned), the limited ability of the models used in the LAMACLIMA project to represent such nuances of land-use practices led to the conclusion that reaching these objectives would imply that all developed scenarios have a similar land-use trajectory over the OECD90 + EU countries. This eventually resulted in the identification of two contrasting scenario narratives: one in which the whole world would follow a path towards sustainable development enabling the achievement of climate and environmental objectives (referred to as *Global-Sustainability*); and one characterized by widening inequalities between the OECD90 + EU countries over which emphasis is put on achieving sustainability objectives locally, and all other countries where these objectives are not met (referred to as *Global-Inequality*).

The final step of the scenario co-development workshop consisted in reviewing the list of relevant processes that can be represented by MAgPIE, and identifying the corresponding parameters or options that should be selected for each scenario and country grouping to best match the underlying scenario narrative. This step was conducted jointly by consortium scientists and stakeholders, and resulted in a concrete list of choices that helped refine their narratives as well as laid the ground for the simulation of both scenarios described in this study.

The scenario co-development process was concluded by an additional online meeting organized half a year after the scenario co-development workshop, which allowed to share first results of the scenario simulations obtained using MAgPIE with the group of stakeholders and validation that the protocol followed when conducting the simulations matched the choices made jointly during the workshop.

### Scenario setup

*Global-Inequality* and *Global-Sustainability* are both scenarios for the AFOLU sector but differ in assumptions for high-income (OECD90 + EU), and low- and middle-income regions (ASIA, LAM, SSA, ROW) with respect to socio-economic development (population, income, diets, food waste), environmental protection (land, water, nitrogen) and land-based mitigation (GHG emission pricing, re/afforestation, bioenergy). The high-income region (OECD90 + EU) includes OECD countries as of 1990 and all EU countries. The setup of the *Global-Sustainability* scenario is comparable to the sustainable development pathway (SDP-1.5 C) presented in Soergel et al.[7]. Details on the scenario setup can be found in Table 4. The SSP storylines underlying the two scenarios have been extended and updated to reflect the outcome of

the scenario co-development process in various aspects such as dietary change, food waste, land protection policies, forestry, and climate change impacts.

Key socio-economic drivers such as population and income are based on SSP4 for *Global-Inequality* and SSP1 for *Global-Sustainability*[13,15,48]. SSP1 is characterized by a lower global population compared to SSP4 after 2050, dominated by different population dynamics in Sub-Saharan Africa in the second half of the 21st century (Fig. 1, Fig. S4). Per-capita income in OECD90 + EU increases similarly in SSP1 and SSP4 over time. In SSP1, the per-capita income of all other regions increases over time as well. In contrast, the gap in per-capita income between low-, middle- and high-income regions widens over time in SSP4, thus increasing global inequality (Fig. 1, Fig. S4). *Global-Inequality* is characterized by resource-intensive diets based on SSP4 with increasing per-capita livestock consumption in all world regions (Figs. 1, 2, Fig. S5). In contrast, diets in *Global-Sustainability* transition to EAT-Lancet recommendations for a healthy diet by 2050, which entails a reduction of livestock products (especially ruminant meat) in favor of fruits, vegetables, nuts, and legumes[7,22]. Calorie overconsumption is gradually reduced over time, resulting in a decline of per-capita consumption in all world regions in *Global-Sustainability*, except for Sub-Saharan Africa where a rapid increase is needed to reduce the prevalence of underweight (Fig. 1). Food waste amounts to 20-25% of total per-capita food supply in 2020 in high- and middle-income regions (Fig. S3). In *Global-Inequality*, food waste shares largely remain within this range over the course of the 21st century in high- and middle-income regions, while food waste shares increase in Sub-Saharan Africa. In *Global-Sustainability*, food waste shares are limited to 20% in all regions in the long term. The combination of higher population and higher per-capita livestock consumption in *Global-Inequality* results in considerably higher total demand for crops (including food and feed crops) and livestock products compared to *Global-Sustainability* (Fig. S6). These socio-economic trajectories are internally consistent regional representations of the underlying storylines. For instance, population and income in high-income regions are similar in both scenarios but differ considerably in low- and middle-income regions. There, no further regional differentiation for the scenario definition is prescribed.

The category of environmental protection targets the impacts of agricultural production on land and water resources (Table 3). Strict protection of forests and other natural lands in all regions in *Global-Inequality* follows the International Union for Conservation of Nature (IUCN) categories I and II of the World Database on Protected Areas (WDPA). Forest and other natural lands in Intact Forest Landscapes (IFL)[49] and Biodiversity Hotspot (BH) areas[50] are protected additionally in OECD90 + EU, and in all regions in *Global-Sustainability*. Strict protection of forests and other natural land areas is ramped up until 2030 and amounts to 767 Mha in *Global-Inequality* and 2522 Mha in *Global-Sustainability* at the global level (see Table S2 for regional numbers). To maintain the resilience and long-term productivity of agricultural landscapes and to guarantee a stable supply of key regulating nature's contribution to people (NCP), native habitats within croplands are restored to a share of 20% of available cropland in OECD90 + EU in *Global-Inequality*, and in all regions in *Global-Sustainability*, in line with the share suggested by a review of the scientific evidence on the topic[51]. To maintain rivers and water bodies, environmental flows are protected in OECD90 + EU in *Global-Inequality*, and in all regions in *Global-Sustainability*. Assumptions for animal waste management and fertilizer soil uptake efficiency are optimistic in all regions in *Global-Sustainability*, and for OECD90 + EU in *Global-Inequality*.

Land-based mitigation includes a price on AFOLU GHG emissions, re/afforestation, and bioenergy. The AFOLU GHG price is active in all regions in *Global-Sustainability*, but only in OECD90 + EU in *Global-Inequality*. Exogenous afforestation based on targets stated by countries in their NDCs is included in both scenarios. Endogenous afforestation depends on the existence of a $CO_2$ price, which serves as incentive for re/afforestation. Based on a review of Fuss et al.[52], "a feasible, yet ambitious boundary limit for global afforestation" of 500 Mha is imposed in *Global-Sustainability* to avoid excessive re/afforestation, which could increase food prices[53]. Demand for second-generation bioenergy is identical in both scenarios. GHG price and bioenergy demand are taken from the sustainable development pathway (SDP-1.5 C) in Soergel et al.[7] (Fig. S9). In line with these assumptions, we account for the impacts of climate change on crop yields, carbon densities, and water availability derived from LPJmL and consistent with the greenhouse gas emissions scenario RCP 1.9, in which global warming peaks below 2 °C and returns below 1.5 °C by 2100. In addition, we account in this study for climate change impacts on labor productivity in line with RCP 1.9 (Fig. S11), which have been included into the production cost function of MAgPIE[21,54–56].

## Data availability
Supplementary Dataset 1 includes numerical scenario results and scripts for figure generation. Supplementary Dataset 2 includes raw data for all figures in the manuscript. Interactive versions of all figures are available at https://magpiemodel.github.io/showcase/lamaclimascenarios.

## Code availability
The source code for MAgPIE 4.4.0 is openly available at https://github.com/magpiemodel and https://doi.org/10.5281/zenodo.5776306. The model documentation can be found at https://rse.pik-potsdam.de/doc/magpie/4.4.0/. Instructions for software installation and running the model are available at https://github.com/magpiemodel. The scenarios have been produced using the script "scripts/start/projects/project_LAMACLIMA_WP4.R".

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

## Acknowledgements

F. Humpenöder, A.P., C.F.S., A.O., M.G.W., I.M., J.P., F. Havermann, W.T., F.L., and Q.L. have received funding through the project "LAnd MAnagement for CLImate Mitigation and Adaptation" (LAMACLIMA). LAMACLIMA is part of AXIS, an ERA-NET initiated by JPI Climate and funded by DLR/BMBF (DE, grant no. 01LS1905C), NWO (NL), RCN (NO), and BELSPO (Belgium) with co-funding by the European Union under Horizon 2020 (Grant Agreement reference 776608). F.L. also acknowledges funding from the Netherlands Organization for Scientific Research (project PERSIST: Persistent Summer Extremes, grant 016.Vidi.171.011). P.v.J. was funded through the SHAPE project. SHAPE is part of AXIS, an ERA-NET initiated by JPI Climate and funded by FORMAS (SE), FFG/BMWFW (AT), DLR/BMBF (DE, grant no. 01LS1907A), NWO (NL) and RCN (NO) with co-funding by the European Union (grant no. 776608). We thank Marcus Lindner, Ybele Hoogeveen, Reinhard Prestele, David Lawrence, and Édouard Léopold Davin for their contributions during the scenario co-development process.

## Author contributions

F. Humpenöder designed the study concept and wrote the manuscript with important contributions from A.P., C.F.S., I.M., and Q.L., and under consideration of feedback from A.O., M.G.W., J.P., F. Havermann, W.T., F.L., P.v.J., J.P.D., H.L., I.W.F. Humpenöder also coordinated the extension of the MAgPIE model, produced the scenario results, analyzed the results, and prepared all figures and tables. The scenario storylines including parametrization in MAgPIE have been co-developed by all authors in a stakeholder engagement process, which was led by Q.L. and I.M. M.G.W. helped with data processing.

## Funding

## Competing interests

The authors declare no competing interests.
