## [Peer Review File · Nature Communications]

Overcoming global inequality is critical for land-based mitigation in line with the Paris AgreementREVIEWER COMMENTS

Reviewer #1 (Remarks to the Author):

General Comments

This study focuses on comparison of contrasting pathways based on SSP4 (inequality) or SSP1 (sustainability), with a range of scenarios where policies and sustainable development activities are applied either globally or for high-income regions as defined by OECD90+EU. While there has been previous work assessing alternative SSPs that reveals similar patterns in the differences in drivers and emissions between SSP1 and SSP4, this study provides a more direct focus on differences between those specific SSPs and a more detailed exploration within a consistent framework of the differences between outcomes when sustainable development is limited to high-income countries. The paper is well-written and clearly presented, highlighting the importance of model assumptions consistent with alternative SSP pathways in affecting projected land-based mitigation potential. The work appears to be methodologically sound, utilizing well-developed models that have been used in a number of previous studies.

Specific Comments

1. Lines 120-121. What do you assume about technological change across these scenarios? Are there differences in productivity growth over time, particularly for low- and middle-income countries that would be expected to have improved productivity growth and more convergence with high-income countries under SSP1 vs. SSP4?
2. Lines 196-202. What about non-CO2 emissions from rice cultivation and manure management? Are they being included here? There is reference to anaerobic digesters and improved water management as supply side technical options, implying that mitigation from those sources is being reflected. Changes in demand for crop and livestock products should be contributing to reducing emissions from these sources from the demand side as well, though.
3. Lines 266-268. What is the pathway in these scenarios through which inclusive socio-economic development is leading to much larger net forest cover increases? Seems like more inclusive development could lead to greater demand for wood products in low- and middle-income regions compared to SSP4, though I imagine there are agricultural and forest productivity gains that are reducing pressure on land resources.
4. Lines 443-449. Are you accounting for carbon sequestration in wood products?
5. Lines 605-607. How was the global limit for re/afforestation of 500 Mha determined?

6. Lines 612-614. How do climate change impacts on labor productivity vary across regions?
7. Line 618. For food waste under SustDemand, is food waste falling from baseline levels (e.g., ~30% in OECD) to a single rate of food waste of 20% in all regions or falling by 20% relative to each regional baseline?
8. Extended Data Figure 1. It appears there is no sugar crop intake showing up in any of the regions. Is it there and very small or being consumed under secondary products: sugar?

Reviewer #2 (Remarks to the Author):

Dear authors, the manuscript deals with a very complex and highly relevant subject, being a work of great socio-environmental impact.

Introduction, results and discussion and methodology are very well structured and written, with no need for modifications.

I have a single suggestion for the first paragraph of the Introduction:

Since the reference "Nabuurs, et al., 2022" is cited it is cited 5 times in the first paragraph of the introduction, I suggest citing the author's name at least in the first citation, for example: "In the period 2010-2019, global GHG emissions from the land sector (AFOLU) accounted for about 21% of global total net anthropogenic GHG emissions, as reported by Nabuurs, et al.1.

After including this small suggestion, and including the suggestions of the other reviewers, I believe that the manuscript will be ready to be published.

Reviewer #3 (Remarks to the Author):

The work deals with a very relevant question investigating how global inequality can affect GHG mitigation options in the AFOLU sector and the synergies and trade-offs with other SDGs. It shows and quantifies striking differences of impacts between a 'global-inequality' and a 'global-sustainability' (less inequal) scenario, as well as the main components of these differences. The global-inequality scenario is far from 1.5° and net-zero pathways by 2100, while global-sustainability scenario is compatible with them, driven mainly by inclusive socio-economic development and GHG emission pricing. A detailed and comprehensive set of results is present together with many insights on mitigation options. I did not find

evidence of flaws and the results shown support the main conclusions. Therefore, the work is recommended to be accepted, with minor revision, according to the following comments:

1. The objective of the study can be understood based on lines 81-89 but it is diffused across the sentences. I suggest a more straightforward description of the objective.
2. As the main figures and tables display the results of the 'five scenarios' (the two main scenarios + the three specific domains) it would be helpful to briefly contextualize them in the objective also.
3. Figure 4 is presented in a different structure from Figures 2 and 3. To facilitate comprehension, I suggest keeping the same structure among them.
4. Figure 5 presents an excellent summary of the overall results. My suggestion is to consider presenting it as the first figure of the results, to give the reader an overview of them before going deep into each specific category of impact. If this option is accepted, the order in which the results are presented must also be adjusted accordingly (subsection 'Synergies...' would be the first in the 'Results' section).
5. Lines 157-159: from Figure 2b, it seems that Asia and Row are relatively as affected by the scenario Global-EnvirProt as LAM and SSA. Please clarify.
6. Lines 182-183: it seems CO2 removals also play an important role on net emissions. Please clarify.
7. Lines 183-185: it seems Asia (middle-income) contributes the most for N2O and CH4 emissions. Please revise the sentence 'especially in low-income regions'.
8. Lines 204-205: it seems LAM also shows considerable potential. Please revise the sentence 'especially in Asia and SSA'.
9. Lines 359-363: this sentence is not supported by the results presented, as the impacts of intermediate scenarios are not shown for SDGs 2 and 4 and as the Global-GHGPrice scenario presents the co-benefit of Forest area increase. Please revise.
10. Please revise abstract conclusions to better reflect the overall results and the above issues. For example, the Global-Sustainability scenario also present substantial effects in LAM and RoW regions AFOLU GHG mitigation and SDGs.

Minor issues:

11. Lines 43-44: IPCC reports AFOLU accounts for 13-21% of global net GHG emissions. I suggest adding the range.
12. Lines 46-47: if possible, please add the percentage of the contribution of 'high' versus 'middle + low-income' countries to the AFOLU emissions, to better support the sentence in line 48.
13. In lines 69-71, the references do not mention the Russian invasion of Ukraine. Please revise.
14. In lines 76-77, disaggregate references to be cited in the corresponding part of the sentence. "access to education is a key driver for lowering population (Lutz, 2019), which in turn...."

15. In Figure 5, the percentages of change are missing in boxes 'a' and 'd'. Please revise.
16. Lines 159-160: if possible, substitute the term 'siloes option' for a more common one.
17. Lines 253-257: maybe the two sentences could be merged and reduced in length.

Point-by-Point response for NCOMMS-22-23985-T

We thank the three referees for their thorough and constructive review of our manuscript.

In this point-by-point response (starting on page 2), the original referee comments are re-printed in black font. Authors' answers to the referee comments are indented and are in blue font. Text in **bold** font is text copied from the revised manuscript.

Summary of major changes, all based on the reviewer comments

- More details on AFOLU GHG emissions in introduction
- Revision of abstract and discussion
- Summary figure and section moved from end to begin of results (Figure 5 -> Figure 3)
- Section on "land-use change" extended by information on land-use intensification, based on new Figure 5.
- Added Figure 7, showing a breakdown of AFOLU GHG emissions by source

Additional changes

- Extended data tables and figures, which are not used in Nature Communications, have been integrated into the main text and methods or moved to the Supplementary Information. The original submission had been prepared for Nature Climate Change, where extended data items are used.
 - Extended Data Table 1 -> Table 2
 - Extended Data Figure 1 -> Figure S3
 - Extended Data Figure 2 -> Figure 2 (now including food waste share)
 - Extended Data Table 2 -> Table S1
 - Extended Data Table 3 -> Table S2
- In total we now have 10 display items: 8 figures and 2 tables.

Begin of point-by-point response

Reviewer #1 (Remarks to the Author):

General Comments

This study focuses on comparison of contrasting pathways based on SSP4 (inequality) or SSP1 (sustainability), with a range of scenarios where policies and sustainable development activities are applied either globally or for high-income regions as defined by OECD90+EU. While there has been previous work assessing alternative SSPs that reveals similar patterns in the differences in drivers and emissions between SSP1 and SSP4, this study provides a more direct focus on differences between those specific SSPs and a more detailed exploration within a consistent framework of the differences between outcomes when sustainable development is limited to high-income countries. The paper is well-written and clearly presented, highlighting the importance of model assumptions consistent with alternative SSP pathways in affecting projected land-based mitigation potential. The work appears to be methodologically sound, utilizing well-developed models that have been used in a number of previous studies.

Specific Comments

1. Lines 120-121. What do you assume about technological change across these scenarios? Are there differences in productivity growth over time, particularly for low- and middle-income countries that would be expected to have improved productivity growth and more convergence with high-income countries under SSP1 vs. SSP4?

Yield-increasing technological change is implemented as endogenous process in MAGPIE. Therefore, our scenarios don't include exogenous assumptions about technological change rates. Instead, the trajectory of yield-increasing technological change is a scenario-specific model output. For each scenario, we calculate an output-related land-use intensity factor τ , which reflects the degree of crop yield amplification caused by human activities.

For clarification we extended the model description in the methods section.
L503-509:

Crop yield increases due to technological change are modeled endogenously based on regionally different investment-yield ratios and interest rates³⁸. Hence, the model simultaneously optimizes the rate of yield-increasing technological change and cropland expansion, which is especially relevant for long-term projections. As indicator for yield-increasing technological change, we calculate the output-related land-use intensity factor τ , which reflects the degree of crop yield amplification caused by human activities³⁹.

38. Dietrich, J. P., Schmitz, C., Lotze-Campen, H., Popp, A. & Müller, C. Forecasting technological change in agriculture—An endogenous implementation in a global land use model. *Technol. Forecast. Soc. Change* **81**, 236–249 (2014).

39. Dietrich, J. P. *et al.* Measuring agricultural land-use intensity – A global analysis using a model-assisted approach. *Ecol. Model.* **232**, 109–118 (2012).

Moreover, we expanded and restructured the whole sub-section on "Land-use change" such that information on land-use intensification is included for all scenarios, based on the newly added Figure 5, which shows the land-use intensity factor τ for each scenario and region.

We renamed the sub-section to "Land-use change and intensification"
For validation with historical data, we added Figure S10 in the SI.

L235-280:

Land-use change and intensification

Global and regional land-use change between 2020 and 2100 differs substantially between *Global-Inequality* and *Global-Sustainability* (Figure 4a). In *Global-Inequality*, expansion of cropland and bioenergy area account for about half of total global land-use expansion by 2100, and timber plantations and afforestation for the other half. About half of total global land-use expansion relies on the reduction of pasture areas, which is facilitated by a shift from pasture-based towards cropland-based animal feed in combination with increased pasture and livestock productivity over time. The other half relies on the conversion of potentially carbon-rich ecosystems such as primary and secondary forest, as well as non-forest natural land. These global developments are driven by heterogeneous regional dynamics (Figure 4b). About 80% of global loss of primary forest, secondary forest, and non-forest natural land between 2020 and 2100 occurs in Sub-Saharan Africa, while there is no loss in high-income regions, the only ones with a price on GHG emissions in *Global-Inequality*. In addition, the GHG price in high-income regions triggers endogenous afforestation with the goal of carbon dioxide removal (see methods). Prescribed afforestation according to nationally determined contributions (NDCs) is especially high in China. Timber plantations for wood production play an important role in high-income regions and Asia, while wood production in other regions largely relies on harvests from natural forests. Land-use intensification, an indicator for human-induced crop yield amplification due to technological change, increases in all regions over time, with higher growth rates in low- and middle-income regions compared to high-income regions (Figure 5).

Global sustainable land-use practices in the scenario *Global-EnvirProt* strongly reduce the loss of primary forest in Sub-Saharan Africa and Latin America, which, however, comes at the cost of more conversion of secondary forest. Besides this difference, overall land-use dynamics as well as land-use intensification in *Global-EnvirProt* are comparable to *Global-Inequality* (Figure 4, Figure 5).

A globally coordinated price on AFOLU GHG emissions in the scenario *Global-GHGPrice* strongly reduces deforestation and conversion of non-forest natural land in all regions. At the same time, the global GHG price shifts re/afforestation from Europe and Northern America to Latin America, which has a higher potential for carbon sequestration. To compensate for reduced land conversion, there is a shift away from cropland expansion towards higher land-use intensification for supplying the same food, feed, bioenergy, and timber as in *Global-Inequality* (Figure 4, Figure 5).

Global inclusive socio-economic development in the *Global-SustDemand* scenario strongly reduces deforestation and conversion of non-forest natural land in all regions, comparable to *Global-GHGPrice*. The declining crop and livestock demand (Figure 1) also frees-up land for more re/afforestation in high-income regions (the only ones with a GHG price in this scenario). At the same time land-use intensification is lower compared to *Global-Inequality* due to the reduced pressure in the land use system (Figure 4, Figure 5).

In the combined scenario, *Global-Sustainability*, declining food demand and global GHG emission pricing interact, which virtually brings deforestation to halt in all regions, increases non-forest natural land in all regions and increases re/afforestation mostly in Latin America. At the same time, land-use intensification is comparable to the relatively low rates of *Global-SustDemand* (i.e. lower than in *Global-Inequality*) in all regions except Asia, where the land-limiting effect of GHG emission pricing results in higher rates of land-use intensification (Figure 4b, Figure 5).

Figure 5: Land-use intensity factor τ . Data is shown at regional level for five scenarios. The τ factor reflects the degree of crop yield amplification caused by human activities. A duplication of τ implies a doubling of crop yields under fixed environmental conditions. See Figure S10 for validation data.

Figure S10: Land-use intensity factor τ . The τ factor reflects the degree of crop yield amplification caused by human activities. A duplication of τ implies a doubling of crop yields under fixed environmental conditions. Data is shown at regional level for the two main scenarios. Historical data for comparison from Dietrich et al 2012 (<https://doi.org/10.1016/j.ecolmodel.2012.03.002>). The historical data has been processed using the pik-piam/mrvalidation R package (<https://doi.org/10.5281/zenodo.4317826>)

2. Lines 196-202. What about non-CO₂ emissions from rice cultivation and manure management? Are they being included here? There is reference to anaerobic digesters and improved water management as supply side technical options, implying that mitigation from those sources is being reflected. Changes in demand for crop and livestock products should be contributing to reducing emissions from these sources from the demand side as well, though.

Yes, non-CO₂ emissions from rice cultivation and manure management are included. For clarification, we revised the corresponding paragraph, see below. In the revised paragraph, we first introduce all non-CO₂ emissions that are included in our modelling framework, followed by the full list of technical mitigation options considered. For more details on AFOLU GHG emissions, we added Figure 7, which shows a breakdown of AFOLU GHG emissions by source.

L315-331

The reduction of AFOLU GHG emissions in *Global-Sustainability* is facilitated by two major factors, one on the demand side (*Global-SustDemand*) and one on the supply side (*Global-GHGPrice*). On the demand side, lower population growth, a transition to healthy diets and reduced food waste lower the demand for crops and livestock products (Figure 1). In turn, these developments a) strongly reduce CO₂ emissions from land conversion and wood harvest, b) increase carbon uptake from re/afforestation, and c) reduce non-CO₂ emissions from agriculture, including N₂O emissions from agricultural soils (fertilizer application) and animal waste management as well as CH₄ emissions from enteric fermentation, animal waste management and rice cultivation (Figure 7). On the supply side, the global GHG price a) strongly reduces the conversion of carbon-rich forests and other ecosystems, b) increases carbon sequestration in managed forests (re/afforestation and timber plantations), and c) reduces CO₂ emissions from managed peatlands through rewetting (Figure 7). Moreover, the GHG price activates the following technical mitigation options for non-CO₂ emissions: 1) changes in animal feed for reducing CH₄ emissions from enteric fermentation, 2) anaerobic digesters for reducing CH₄ and N₂O emissions from animal waste management, 3) improved water management for reducing CH₄ emissions from rice cultivation and 4) improved fertilizer application for reducing N₂O emissions from agricultural soils.

3. Lines 266-268. What is the pathway in these scenarios through which inclusive socio-economic development is leading to much larger net forest cover increases? Seems like more inclusive development could lead to greater demand for wood products in low- and middle-income regions compared to SSP4, though I imagine there are agricultural and forest productivity gains that are reducing pressure on land resources.

Inclusive socio-economic development (*Global-SustDemand*) lowers total crop and livestock demand and hence frees up land resources for re/afforestation. In consequence, re/afforestation area in OECD90+EU is about twice as high as in the *Global-inequality* scenario. In combination with reduced deforestation in Latin America and Sub-Saharan Africa this results in higher net forest cover gains. Demand for wood products does affect the indicator we use in Figure 5d (now Figure 3d) because a) forest remains forest after wood harvest and b) timber plantations are excluded from the indicator.

For clarification we revised the corresponding paragraph in the results section.

L400-413

At the global level, net forest cover increases between 2020 and 2050 by 398 Mha in *Global-Sustainability* and 115 Mha in *Global-inequality* (Figure 3d). The higher increase of forest

cover in *Global-Sustainability* is facilitated by two major factors: a) a global GHG price (*Global-GHGPrice*), which provides an economic incentive for forest protection and restoration, and b) inclusive socio-economic development (*Global-SustDemand*), which lowers total crop and livestock demand and hence frees up land resources for re/afforestation. While the outcome of both scenarios is similar at the global level, the regional allocation of re/afforestation differs considerably (Figure 8, Figure 3). Re/afforestation area roughly doubles in OECD90+EU under inclusive socio-economic development compared to *Global-inequality* (Figure 4) due to the combined effects of GHG emission pricing (OECD90+EU has a GHG price in all scenarios) and reduced pressure on land from food production. In contrast, a global GHG price shifts re/afforestation from Europe and Northern America to Latin America, which has a higher potential for terrestrial carbon sequestration.

For further clarification we also revised one sentence in the introduction.
L73-77

In such sustainable transformation pathways, global GHG emission pricing is often complemented with global-scale inclusive socio-economic development (a convergence of all countries towards e.g. the same demographic profiles, lower food waste levels and healthy diets) and sustainable land-use practices (e.g. ecosystem protection and efficiency improvements)^{7,8}.

4. Lines 443-449. Are you accounting for carbon sequestration in wood products?

Yes. For clarification, we added a sentence in the methods section.
L532-534

The calculation of CO₂ emissions from wood harvest, carbon storage in wood products and carbon uptake in timber plantations follows the methodology described in Mishra et al⁴⁰.

40. Mishra, A. *et al.* Estimating global land system impacts of timber plantations using MAgPIE 4.3.5. *Geosci. Model Dev.* 14, 6467–6494 (2021).

In addition, we added Figure 7, which shows a breakdown of AFOLU GHG emissions by source. We refer to Figure 7 in the following sentence.
L319-323

In turn, these developments a) strongly reduce CO₂ emissions from land conversion and wood harvest, b) increase carbon uptake from re/afforestation, and c) reduce non-CO₂ emissions from agriculture, including N₂O emissions from agricultural soils (fertilizer application) and animal waste management as well as CH₄ emissions from enteric fermentation, animal waste management and rice cultivation (Figure 7).

Figure 7: AFOLU GHG emission breakdown by source. Data is shown at global level for five scenarios. a) shows CO₂ emissions and removals from land-use change and management. Carbon losses consist of emissions from deforestation, conversion of non-forest ecosystems, drained peatlands and wood harvest. Carbon gains consist of carbon storage in wood products, re/afforestation (Aff CO₂-price and Aff NDC), timber plantations and regrowth of natural vegetation (secondary forests and other natural land). The black line shows the net effect of carbon losses and carbon gains at global level. b) shows CH₄ emissions from agriculture. c) shows N₂O emissions from agriculture. Further details on AFOLU GHG emission sources and sinks are provided in the methods section and in Table S1.

5. Lines 605-607. How was the global limit for re/afforestation of 500 Mha determined?

We added more details to the corresponding sentence in the methods section. L688-690

Based on a review of Fuss et al⁵³, "a feasible, yet ambitious boundary limit for global afforestation" of 500 Mha is imposed in *Global-Sustainability* to avoid excessive re/afforestation, which could increase food prices⁵⁴.

53. Fuss, S. et al. Negative emissions—Part 2: Costs, potentials and side effects. *Environ. Res. Lett.* **13**, 063002 (2018).

54. Kreidenweis, U. et al. Afforestation to mitigate climate change: impacts on food prices under consideration of albedo effects. *Environ. Res. Lett.* **11**, 085001 (2016).

6. Lines 612-614. How do climate change impacts on labor productivity vary across regions?

Climate change impacts on labor productivity under RCP 1.9 are small in most regions, with exception of South Asia.

For clarification, we added a map in the SI.

Figure S11: Climate change impacts on labor productivity under RCP 1.9. Data is shown as index relative to full labor productivity (1). Heat-induced impacts on labor productivity for RCP 1.9 (ISO metric at 400 watt work intensity) have been calculated using the methodology described in Orlov et al 2021 (<https://doi.org/10.1007/s41885-021-00091-6>), based on data from the LAMACLIMA project.

7. Line 618. For food waste under SustDemand, is food waste falling from baseline levels (e.g., ~30% in OECD) to a single rate of food waste of 20% in all regions or falling by 20% relative to each regional baseline?

Food waste is limited to 20% of food supply in *Global-SustDemand* and *Global-Sustainability*.

For clarification, we added the corresponding food waste shares to Extended Data Figure 2 (now Figure 2) and revised the corresponding sentences in the methods sections.

L652-656

Food waste amounts to 20-25% of total per-capita food supply in 2020 in high- and middle-income regions (Figure S3). In *Global-Inequality*, food waste shares largely remain within this range over the course of the 21st century in high- and middle-income regions, while food waste shares increase in Sub-Saharan Africa. In *Global-Sustainability*, food waste shares are limited to 20% in all regions in the long-term.

We also revised Extended Data Table 1 (now Table 2). Note that the previous number of ~30% food waste in OECD was taken from the literature (Lamb et al 2021 <https://doi.org/10.1088/1748-9326/abee4e>).

Socio-economic drivers and dietary change		Population	Income	Diet	Food Waste (% of food supply)	Timber Demand
SustDemand	off	SSP4	SSP4	SSP4	no constraint	SSP4
	on	SSP1	SSP1	EAT-Lancet	limited to 20%	SSP1

8. Extended Data Figure 1. It appears there is no sugar crop intake showing up in any of the regions. Is it there and very small or being consumed under secondary products: sugar?

Indeed, sugar is only consumed as secondary product.

We revised Extended Data Figure 1 (now Figure S3) accordingly. We also removed the "+" signs from the variable names in the legend.

Figure S3: Per-capita calorie intake. a) shows regional data for 2020. The height of each rectangle shows per-capita kcal intake, the width shows the population of the region, so that the area of the rectangles refers to the total calorie intake for each region. b) shows EAT-Lancet recommendations (planetary health diet), aggregated to global level. In the Global-Sustainability scenario, all regions converge towards the EAT-Lancet recommendation by 2050.

Reviewer #2 (Remarks to the Author):

Dear authors, the manuscript deals with a very complex and highly relevant subject, being a work of great socio-environmental impact.

Introduction, results and discussion and methodology are very well structured and written, with no need for modifications.

I have a single suggestion for the first paragraph of the Introduction:

Since the reference "Nabuurs, et al., 2022" is cited it is cited 5 times in the first paragraph of the introduction, I suggest citing the author's name at least in the first citation, for example: "In the period 2010-2019, global GHG emissions from the land sector (AFOLU) accounted for about 21% of global total net anthropogenic GHG emissions, as reported by Nabuurs, et al.1.

We adopted this suggestion:

L44-46

In the period 2010-2019, global GHG emissions from the land sector (AFOLU) accounted for 13-21% of global total net anthropogenic GHG emissions, according to IPCC AR6 WGIII¹

After including this small suggestion, and including the suggestions of the other reviewers, I believe that the manuscript will be ready to be published.

Reviewer #3 (Remarks to the Author):

The work deals with a very relevant question investigating how global inequality can affect GHG mitigation options in the AFOLU sector and the synergies and trade-offs with other SDGs. It shows and quantifies striking differences of impacts between a 'global-inequality' and a 'global-sustainability' (less unequal) scenario, as well as the main components of these differences. The global-inequality scenario is far from 1.5° and net-zero pathways by 2100, while global-sustainability scenario is compatible with them, driven mainly by inclusive socio-economic development and GHG emission pricing. A detailed and comprehensive set of results is present together with many insights on mitigation options. I did not find evidence of flaws and the results shown support the main conclusions. Therefore, the work is recommended to be accepted, with minor revision, according to the following comments:

1. The objective of the study can be understood based on lines 81-89 but it is diffused across the sentences. I suggest a more straightforward description of the objective.

We revised the corresponding sentences in the introduction.
L95-101

In light of the closing window for reaching the global 1.5°C target¹¹, it seems highly relevant to study how (in)compatible AFOLU GHG emissions in a world of deepening inequalities would be with the Paris Agreement climate objectives⁶. Likewise, a forward-looking analysis of expanding mitigation options from high-income regions to global level can provide insights on the most effective options for AFOLU emission reduction in low- and middle-income regions under consideration of co-benefits and trade-offs with other SDGs.

new reference:

6. Schleussner, C.-F., Ganti, G., Rogelj, J. & Gidden, M. J. An emission pathway classification reflecting the Paris Agreement climate objectives. *Commun. Earth Environ.* **3**, 1–11 (2022).

2. As the main figures and tables display the results of the 'five scenarios' (the two main scenarios + the three specific domains) it would be helpful to briefly contextualize them in the objective also.

We revised the corresponding sentences in the introduction.
L102-111

Here, we present a model-based quantification of two contrasting future scenarios for the AFOLU sector, complemented by three intermediate scenarios (Table 1; see Table 2 and Methods for details). *Global-Inequality* is a scenario where sustainable development in the land sector remains highly unequal and limited to high-income countries only, whereas the scenario *Global-Sustainability* assumes global implementation of AFOLU GHG emission pricing, sustainable land-use practices and inclusive socio-economic development. The two main scenarios are complemented by three intermediate scenarios for decomposing the main drivers for the transformation from *Global-Inequality* towards *Global-Sustainability*: Global AFOLU GHG emission pricing (*Global-GHGPrice*), global sustainable land-use practices (*Global-EnvirProt*) and global inclusive socio-economic development (*Global-SustDemand*).

3. Figure 4 is presented in a different structure from Figures 2 and 3. To facilitate comprehension, I suggest keeping the same structure among them.

The same structure as in Figure 2 (land-use change in Mha) and Figure 3 (GHG emission in Gt CO₂eq) is not possible because the variables shown in Figure 4 have different units. However, we changed Figure 4 such that colors are used for indicating the regions; the same colors as used in Figure 1 and Figure 5.

Note that due to the shifting of Figure 5 (see below) and the integration of Extended Display Items into the main text, Figure 4 has been renamed to Figure 8.

Figure 8: Change in agricultural water use, nitrogen fixation and forest area compared to 2020 for five scenarios at regional level for the years 2050 and 2100. The black dot indicates the net effect at global level.

4. Figure 5 presents an excellent summary of the overall results. My suggestion is to consider presenting it as the first figure of the results, to give the reader an overview of them before going deep into each specific category of impact. If this option is accepted, the order in which the results are presented must also be adjusted accordingly (subsection 'Synergies...' would be the first in the 'Results' section).

Thanks for this suggestion. We moved Figure 5 (now Figure 3) and the corresponding sub-section to the beginning of the results.

We renamed the sub-section to "**Synergies and trade-offs between land-based mitigation and other SDGs**".

We also revised the beginning of the sub-section.
L153-156

This sub-section provides a summary of the main scenario results for the year 2050 with a focus on co-benefits and trade-offs between land-based mitigation and other SDGs. Detailed results on land-use dynamics, AFOLU GHG emissions, agricultural water use, and nitrogen fixation are provided in the subsequent sub-sections.

We also revised the paragraphs on AFOLU GHG emission pricing and inclusive socio-economic development in this sub-section; providing a better overview of the underlying causes for AFOLU GHG emission reduction.

L187-193

The global GHG price has three major effects on net CO₂ emissions from land-use change and management: a) strong reduction of CO₂ emissions from land conversion in Sub-Saharan Africa, Latin America and Asia, b) increased carbon sequestration in managed forests, including a shift of re/afforestation from OECD90+EU to Latin America, and c) reduced CO₂ emissions from drained peatlands through rewetting (Figure 7). These processes result in net global carbon uptake of 4.2 Gt CO₂ eq yr⁻¹ in 2050 (Figure 3).

L202-207

The combined effects of lower population growth, transition to healthy diets and reduced food waste lower the demand for agricultural commodities (Figure 1). In turn, the reduced pressure on limited land resources a) strongly reduces CO₂ emissions from land conversion in Asia, Sub-Saharan Africa and Latin America, b) increase carbon uptake from re/afforestation, and c) reduces non-CO₂ emissions from agriculture in all regions (Figure 3, Figure 7).

5. Lines 157-159: from Figure 2b, it seems that Asia and Row are relatively as affected by the scenario *Global-EnvirProt* as LAM and SSA. Please clarify.

The absolute differences for Asia and ROW between the scenarios *Global-EnvirProt* and *Global-Inequality* are quite small. Therefore, we only mention the larger differences in primary forest for Sub-Saharan Africa and Latin America in the manuscript text.

Note that we revised the whole sub-section on land-use change.
The corresponding sentence reads now as follows.

L256-260

Global sustainable land-use practices in the scenario *Global-EnvirProt* strongly reduce the loss of primary forest in Sub-Saharan Africa and Latin America, which, however, comes at the cost of more conversion of secondary forest. Besides this difference, overall land-use dynamics as well as land-use intensification in *Global-EnvirProt* are comparable to *Global-Inequality* (Figure 4, Figure 5).

6. Lines 182-183: it seems CO₂ removals also play an important role on net emissions. Please clarify.

We revised the corresponding sentence accordingly.
The newly added Extended Data Figure 4 shows a break-down of AFOLU GHG emissions by source.

L299-303

Global net AFOLU GHG emissions remain positive in *Global-Inequality* throughout the 21st century with levels of 8 Gt CO₂ eq yr⁻¹ in 2050 and 5 Gt CO₂ eq yr⁻¹ in 2100. These GHG emissions are largely caused by N₂O and CH₄ emissions from agriculture, while global net-CO₂ emissions from land-use change and management are close to zero (Figure 7).

The reason for close to net-zero CO₂ emissions is explained some sentences later. We slightly adjusted this sentence.

L306-309

Global net CO₂ emissions from land-use change and management are close to zero mostly because of re/afforestation-based carbon sequestration in high-income regions that compensates for increasing CO₂ emissions from deforestation in Sub-Saharan Africa (Figure 6b).

We also changed the terminology to "net CO₂ emissions from land-use change and management" or "CO₂ emissions and removals from land-use change and management" in other places of the manuscript to highlight that CO₂ removals play an important role.

7. Lines 183-185: it seems Asia (middle-income) contributes the most for N₂O and CH₄ emissions. Please revise the sentence 'especially in low-income regions'.

We revised the whole sentence.

L303-306

N₂O and CH₄ emissions decrease in high-income regions but continue or even increase in low- and middle-income regions until 2050. By 2100, N₂O and CH₄ emissions decrease in all regions except Sub-Saharan Africa, mostly due to population dynamics in combination with per-capita livestock product consumption patterns (Figure 1).

8. Lines 204-205: it seems LAM also shows considerable potential. Please revise the sentence 'especially in Asia and SSA'.

We revised the sentence accordingly.

L331-335

Both options, AFOLU GHG emission pricing and inclusive socio-economic development, show considerable potential for AFOLU GHG emission reduction by 2050 and beyond, especially in Asia, Latin America and Sub-Saharan Africa, resulting in net-negative global AFOLU GHG emissions from around 2050 onwards (Figure 6).

9. Lines 359-363: this sentence is not supported by the results presented, as the impacts of intermediate scenarios are not shown for SDGs 2 and 4 and as the Global-GHGPrice scenario presents the co-benefit of Forest area increase. Please revise.

Indeed, the impacts on SDG 2 and SDG 4 are only shown for the two main scenarios in Figure 1. However, by assumption the impacts on SDG 2 and SDG 4 are identical in Global-SustDemand and Global-Sustainability. For clarification we revised the legend of Figure 1 accordingly. We also revised the legend of Extended Data Fig 2 (now Figure 2).

Figure 1: Key socio-economic drivers and developments. Data is shown at regional and global level for the two main scenarios Global-Inequality (same for Global-GHGPrice and Global-EnvirProt) and Global-Sustainability (same for Global-SustDemand). a) Population, b) Per-capita income, c) Per-capita calorie intake, d) Total demand for crops (including food and feed) and livestock products in million-ton dry matter per year, e) Prevalence of underweight (body mass index < 18.5) and f) Prevalence of obesity (BMI > 30). Details on prevalence of underweight and obesity can be found in Table S1.

Figure 2: Per-capita calorie supply. The sum of crops, livestock products, fish and secondary products reflects food intake, which together with food waste sums up to food supply. Percentage values indicate the share of food waste in total food supply. Data is shown at regional level for the two main scenarios Global-Inequality (same for Global-GHGPrice and Global-EnvirProt) and Global-Sustainability (same for Global-SustDemand).

10. Please revise abstract conclusions to better reflect the overall results and the above issues. For example, the Global-Sustainability scenario also present substantial effects in LAM and RoW regions AFOLU GHG mitigation and SDGs.

We revised abstract and discussion/conclusion accordingly.

Abstract
L35-38

While also a scenario purely based on either global GHG emission pricing or on inclusive socio-economic development would achieve the stringent emissions reductions required, only the latter ensures major co-benefits for other Sustainable Development Goals, especially in low- and middle-income regions.

Discussion
L428-432

However, only inclusive socio-economic development comes with multiple co-benefits, especially in low- and middle-income regions, for other SDGs than climate protection (SDG 13), including malnutrition (SDG 2), overnutrition (SDG 4), ecosystem conservation and restoration (SDG 15), nitrogen pollution (SDG 15) and agricultural water use (SDG 6).

Minor issues:

11. Lines 43-44: IPCC reports AFOLU accounts for 13-21% of global net GHG emissions. I suggest adding the range.

We added the range as suggested
L44-46

In the period 2010-2019, global GHG emissions from the land sector (AFOLU) accounted for 13-21% of global total net anthropogenic GHG emissions, according to IPCC AR6 WGIII¹.

In addition, we added more details on AFOLU GHG emissions in the introduction.
L46-53

GHG emissions from AFOLU consist of two major components: a) carbon dioxide (CO₂) emissions and removals from land-use change and management, and b) methane (CH₄) and nitrous oxide (N₂O) emissions from agriculture². In 2018, CO₂ emissions from deforestation and other land conversions together with carbon uptake due to regrowth and re/afforestation accounted for 47% of global net AFOLU GHG emissions, followed by CH₄ emissions from enteric fermentation with 25%². Smaller sources of AFOLU GHG emissions include managed soils (N₂O), rice cultivation (CH₄), manure management (CH₄ and N₂O) and synthetic fertilizer application (N₂O).

12. Lines 46-47: if possible, please add the percentage of the contribution of 'high' versus 'middle + low-income' countries to the AFOLU emissions, to better support the sentence in line 48.

We added more relative and absolute numbers on regional AFOLU GHG emissions, based on two additional references.

L53-60

Unlike other sectors such as energy, industry and transport, the relative contribution of AFOLU to overall GHG emissions is typically higher in developing than in developed countries². In Africa, Latin America and South East Asia, AFOLU GHG emissions account for more than 50% of total GHG emissions, mostly due to high CO₂ emissions from land-use change and management^{1,2}. In contrast, the share of AFOLU in total GHG emissions is only 7% each in Europe and North America^{1,2}. In absolute terms, AFOLU GHG emissions in 2015 were considerably higher in developing countries (9.5 Gt CO₂ eq) compared to industrialized countries (2.6 Gt CO₂ eq)³.

2. Lamb, W. F. *et al.* A review of trends and drivers of greenhouse gas emissions by sector from 1990 to 2018. *Environ. Res. Lett.* 16, 073005 (2021).

3. Crippa, M. *et al.* Food systems are responsible for a third of global anthropogenic GHG emissions. *Nat. Food* 1–12 (2021) doi:10.1038/s43016-021-00225-9.

13. In lines 69-71, the references do not mention the Russian invasion of Ukraine. Please revise.

We revised the corresponding sentences and added a reference for the possible impacts of the Russian invasion of Ukraine. Moreover, we shifted these sentences towards the end of the paragraph to avoid that the discussion of barriers is interrupted.

L91-95

Also the impacts of the COVID-19 pandemic on the global economy will most likely further slow down progress across SDGs¹⁶, especially in low-income countries in Sub-Saharan Africa¹⁷. Moreover, the Russian invasion of Ukraine might negatively affect food security, and in consequence political stability, in countries with limited coping capacity, especially in Middle East and Africa¹⁸.

16. Naidoo, R. & Fisher, B. Reset Sustainable Development Goals for a pandemic world. *Nature* 583, 198–201 (2020).

17. Josephson, A., Kilic, T. & Michler, J. D. Socioeconomic impacts of COVID-19 in low-income countries. *Nat. Hum. Behav.* 5, 557–565 (2021).

18. Hellegers, P. Food security vulnerability due to trade dependencies on Russia and Ukraine. *Food Secur.* (2022) doi:10.1007/s12571-022-01306-8.

14. In lines 76-77, disaggregate references to be cited in the corresponding part of the sentence. "access to education is a key driver for lowering population (Lutz, 2019), which in turn...."

We revised the sentence accordingly.

L86-88

For instance, access to education is a key driver for lowering population growth¹², which in turn is a key driver for future food demand and agricultural land use^{13,14}.

15. In Figure 5, the percentages of change are missing in boxes 'a' and 'd'. Please revise.

Unfortunately, this is not possible. a) and d) in Figure 5 (now Figure 3) show indicators that can turn net-zero or net-negative at global level. Therefore, a meaningful global percentage change compared to the *Global-Inequality* scenario is not possible in all cases. For instance, global annual net CO₂ emissions are close to zero in the *Global-Inequality* scenario. Using a value close to zero as denominator would not produce meaningful percentage changes.

For clarification, we revised and expanded the caption of Figure 3.

Figure 3: Summary of results for five scenarios and six indicators (mapped to SDGs) for 2050 at regional and global level. a) CO₂ emissions and removals from land-use change and management, b) N₂O emissions from agriculture, c) CH₄ emissions from agriculture, d) change in forest area without plantations, e) nitrogen fixation and f) agricultural water use. Black dots show the global net effect for each indicator and scenario. Blue vertical lines show the respective value in year 2020. Percentage labels in panels b, c, e and f show the relative change of the global indicator level for each scenario compared to the Global-Inequality scenario. For panels a and d it is not meaningful to show percentage changes because these indicators can, by definition, turn net-zero (in case of divergent regional dynamics) or net-negative at global level.

16. Lines 159-160: if possible, substitute the term 'siloe option' for a more common one.

We revised the sentence.
L256-258

Global sustainable land-use practices in the scenario *Global-EnvirProt* strongly reduce the loss of primary forest in Sub-Saharan Africa and Latin America, which, however, comes at the cost of more conversion of secondary forest.

17. Lines 253-257: maybe the two sentences could be merged and reduced in length.

We shortened and merged the two sentences.
L390-393

The reduction of nitrogen fixation in *Global-Sustainability* is facilitated by inclusive socio-economic development (*Global-SustDemand*) and sustainable land-use practices (*Global-EnvirProt*), which reduce nitrogen fixation by 26% and 23%, respectively (Figure 3).

REVIEWERS' COMMENTS

Reviewer #1 (Remarks to the Author):

The authors were very thorough and did an excellent job in addressing each of my review comments on the original submission. I do not have additional comments.

Reviewer #2 (Remarks to the Author):

The authors answered the reviewers' questions in detail, having responded to the suggestions received.

The work has a solid scientific and methodological basis, and its results are very well analyzed and discussed. I believe the article is ready for publication.

Reviewer #3 (Remarks to the Author):

The authors have revised the paper and either resolved or justified each of the issues or suggestions presented in the review. In my view, the revised version is clearer and the new text and figures presented improve the consistency and reproducibility of the work as well as making some results easier to understand. I reiterate my main perceptions in the first round of review and suggest the paper is suitable for publication. I congratulate the authors for the excellence of the study and thank them for the opportunity to learn from it.

Regarding land-use intensity, I would like to suggest a recent document on Brazil's agricultural land area, in case it can be useful in future improvements of the models.

Novaes, R.M.L., Tubiello, F.N., Garofalo, D.F.T., Santis, G., Pazianotto, R.A., Matsuura, M.I.S.F., 2022. Brazil's Agricultural Land, Cropping Frequency and Second Crop Area: FAOSTAT Statistics and New Estimates. Embrapa Meio Ambiente, Jaguariúna, Brazil.

Kind regards,

Renan Novaes

Point-by-Point response for NCOMMS-22-23985B

We thank the three referees for their thorough and constructive review of our manuscript.

In this point-by-point response (starting on page 2), the original referee comments are re-printed in black font. Authors' answers to the referee comments are indented and are in blue font. Text in **bold** font is text copied from the revised manuscript.

Begin of point-by-point response

Reviewer #1 (Remarks to the Author):

The authors were very thorough and did an excellent job in addressing each of my review comments on the original submission. I do not have additional comments.

We appreciate your positive feedback and thank you again for reviewing our manuscript.

Reviewer #2 (Remarks to the Author):

The authors answered the reviewers' questions in detail, having responded to the suggestions received.

The work has a solid scientific and methodological basis, and its results are very well analyzed and discussed. I believe the article is ready for publication.

We appreciate your positive feedback and thank you again for reviewing our manuscript.

Reviewer #3 (Remarks to the Author):

The authors have revised the paper and either resolved or justified each of the issues or suggestions presented in the review. In my view, the revised version is clearer and the new text and figures presented improve the consistency and reproducibility of the work as well as making some results easier to understand. I reiterate my main perceptions in the first round of review and suggest the paper is suitable for publication. I congratulate the authors for the excellence of the study and thank them for the opportunity to learn from it.

Regarding land-use intensity, I would like to suggest a recent document on Brazil's agricultural land area, in case it can be useful in future improvements of the models.

Novaes, R.M.L., Tubiello, F.N., Garofalo, D.F.T., Santis, G., Pazianotto, R.A., Matsuura, M.I.S.F., 2022. Brazil's Agricultural Land, Cropping Frequency and Second Crop Area: FAOSTAT Statistics and New Estimates. Embrapa Meio Ambiente, Jaguariúna, Brazil.

Kind regards,
Renan Novaes

We appreciate your positive feedback and thank you again for reviewing our manuscript. We will consider the study on land-use intensity in Brazil in future model development.